# Unconventional secretion of α-synuclein mediated by palmitoylated DNAJC5 oligomers

**Shenjie Wu[1], Nancy C Hernandez Villegas[2], Daniel W Sirkis[3], Iona Thomas-Wright[4], Richard Wade-Martins[4], Randy Schekman[1]***

[1]Department of Molecular and Cell Biology, Howard Hughes Medical Institute, University of California, Berkeley, Berkeley, United States; [2]Helen Wills Neuroscience Institute, University of California, Berkeley, Berkeley, United States; [3]Memory and Aging Center, Department of Neurology, University of California, San Francisco, San Francisco, United States; [4]Oxford Parkinson's Disease Centre, Department of Physiology, Anatomy and Genetics and Kavli Institute for Nanoscience Discovery, University of Oxford, Oxford, United Kingdom

**Abstract** Alpha-synuclein (α-syn), a major component of Lewy bodies found in Parkinson's disease (PD) patients, has been found exported outside of cells and may mediate its toxicity via cell-to-cell transmission. Here, we reconstituted soluble, monomeric α-syn secretion by the expression of DnaJ homolog subfamily C member 5 (DNAJC5) in HEK293T cells. DNAJC5 undergoes palmitoylation and anchors on the membrane. Palmitoylation is essential for DNAJC5-induced α-syn secretion, and the secretion is not limited by substrate size or unfolding. Cytosolic α-syn is actively translocated and sequestered in an endosomal membrane compartment in a DNAJC5-dependent manner. Reduction of α-syn secretion caused by a palmitoylation-deficient mutation in DNAJC5 can be reversed by a membrane-targeting peptide fusion-induced oligomerization of DNAJC5. The secretion of endogenous α-syn mediated by DNAJC5 is also found in a human neuroblastoma cell line, SH-SY5Y, differentiated into neurons in the presence of retinoic acid, and in human-induced pluripotent stem cell-derived midbrain dopamine neurons. We propose that DNAJC5 forms a palmitoylated oligomer to accommodate and export α-syn.

***For correspondence:**
schekman@berkeley.edu

## Editor's evaluation

This study describes a comprehensive set of classical techniques in biochemistry and cell biology characterizing the unconventional secretory mechanism of α-synuclein, an important contributor to neurodegeneration. The major finding is that palmitoylated DNAJC5 oligomers play a central role in this unusual secretory pathway, presumably by mediating entry of α-synuclein into late endosomes. Future work will clarify the mechanisms by which this cargo is exported from the cell.

## Introduction

Parkinson's disease (PD), the second most common neurodegenerative disease, is characterized by the deposit of clumps of protein aggregate, lipid and damaged organelles known as Lewy bodies (LBs) (*Dauer and Przedborski, 2003*; *Shahmoradian et al., 2019*). One of the main constituents of LB is the presynaptic protein alpha-synuclein (α-syn) (*Stefanis, 2012*). α-syn is encoded by the *SNCA* gene and is highly abundant in neurons. As a small, intrinsically disordered protein-containing 140 amino acids (AAs), α-syn can be divided into three domains, an amphipathic N-terminal domain where

**Figure 1.** Reconstitution of α-syn secretion regulated by palmitoylated DNAJC5 in HEK293T cells. (**A**) Schematic diagrams of α-syn and DNAJC5. Domains are highlighted in different colors. Red arrows indicate known disease-causing mutations on each protein. (**B**) Membrane and cytosol fractionation scheme. Briefly, homogenized HEK293T cells were centrifuged at low speed to prepare a post-nuclear supernatant (PNS). High-speed centrifugation was then performed to separate the sedimentable membrane (M) from cytosol (C). (**C**) Partition of palmitoylated DNAJC5 (P-DNAJC5) and non-palmitoylated DNAJC5 (NP-DNAJC5) between the membrane (M) and cytosol (C) fractions. DNAJC5 was immunoprecipitated from cytosol and membrane with anti-FLAG resin and evaluated by Coomassie-blue stained SDS-PAGE. (**D**) α-syn secretion 16 h after transfection. The secretion of P-DNAJC5 in the medium was detected. (**E**) α-syn secretion 36 h after transfection. NP-DNAJC5 was also secreted in the medium together with α-syn.

The online version of this article includes the following source data and figure supplement(s) for figure 1:

**Source data 1.** Uncropped immunoblot and gel images corresponding to *Figure 1*.

**Figure supplement 1.** Validation of palmitoylation of DNAJC5 in various cell lines.

**Figure supplement 1—source data 1.** Uncropped immunoblot images corresponding to *Figure 1—figure supplement 1*.

**Figure supplement 2.** Secretion of α-syn variants.

**Figure supplement 2—source data 1.** Uncropped immunoblot images corresponding to *Figure 1—figure supplement 2*.

**Figure supplement 3.** Secretion of α-syn is partially dependent on endogenous DNAJC5 in HEK293T cells.

**Figure supplement 3—source data 1.** Uncropped immunoblot images corresponding to *Figure 1—figure supplement 3*.

most PD-related mutations are located, including A30P, E46K, and A53T, a central hydrophobic region known as the non-amyloid-β component (NAC) which is essential for aggregation, and an acidic C-terminal domain (*Figure 1A*; *Alderson and Markley, 2013*). α-syn can undergo a conformational change from a disordered monomer to an oligomer (*Burré et al., 2014*; *Lashuel et al., 2002*), which can further polymerize to form insoluble fibrils (*Guerrero-Ferreira et al., 2019*; *Guerrero-Ferreira et al., 2018*; *Strohäker et al., 2019*).

In recent years, studies have suggested that α-syn deposits are not static, but rather actively spread during disease progression. Grafted neurons in PD patients developed α-syn positive LBs years after surgery, suggesting host-to-graft pathology propagation (*Kordower et al., 2008*; *Li et al., 2008*). Based on analysis of human pathology, the Braak hypothesis posits that α-syn aggregates can spread in a stereotyped manner from the gastrointestinal tract to the brain, causing neuron loss beginning in

the brainstem, extending to the midbrain, and finally to the cortex (*Braak et al., 2006*; *Braak et al., 2003*). In more recent work, Braak-like transmission of in vitro generated α-syn fibrils has been recapitulated in mice and non-human primates (*Chu et al., 2019*; *Chung et al., 2020*; *Kim et al., 2019*; *Luk et al., 2012*).

Less-well understood is the molecular and cellular basis for the transfer of α-syn between cells. Several mechanisms have been proposed for both monomeric and aggregated α-syn export and transfer, including unconventional exocytosis (*Jang et al., 2010*; *Lee et al., 2005*), exosomes (*Danzer et al., 2012*; *Emmanouilidou et al., 2010*; *Stykel et al., 2021*), and membrane nanotubes (*Abounit et al., 2016*; *Scheiblich et al., 2021*). The extracellular existence of both monomeric and oligomeric α-syn was confirmed in blood and cerebrospinal fluid (CSF) (*Borghi et al., 2000*; *El-Agnaf et al., 2003*). Packaging of different conformational forms of α-syn inside extracellular vesicles (EVs) has been reported but requires further vigorous scrutiny to differentiate membrane vesicles from secreted sedimentable aggregates (*Brahic et al., 2016*).

DNAJC5, also known as cysteine string protein α (CSPα), is a co-chaperone of HSC70 and has been shown to control the extracellular release of many neurodegenerative disease proteins (*Fontaine et al., 2016*). This process of unconventional traffic has been termed misfolding-associated protein secretion (MAPS) (*Fontaine et al., 2016*; *Lee et al., 2016*) as opposed to conventional secretion initiated by an amino-terminal signal peptide required for secretory and membrane protein translocation into the endoplasmic reticulum (*Zhang and Schekman, 2013*). DNAJC5 contains three domains—a common N-terminal J-domain conserved among DnaJ proteins, the cysteine-string (CS) central domain which is heavily palmitoylated and anchors the protein to late endosomes, and an overall disordered C-terminal domain (*Figure 1A*). Deletion of DNAJC5 in *Drosophila* and mice leads to a neurodegenerative phenotype and premature death, indicating that DNAJC5 plays a neuroprotective role in the brain (*Zinsmaier, 2010*). Transgenic expression of α-syn appears to rescue the neurodegeneration seen on depletion of DNAJC5 (*Chandra et al., 2005*). A previous study also reported that neuron-derived EVs contain DNAJC5 (*Deng et al., 2017*). However, the mechanism by which DNAJC5 recognizes and translocates soluble α-syn into a membrane compartment for secretion remains elusive.

In this study, we characterized the mechanism of DNAJC5-induced α-syn secretion in a cell-based secretion assay. Using biochemical characterization and imaging of internalized α-syn in enlarged endosomes as a secretory intermediate, we found previously underappreciated roles of palmitoylation and oligomerization of DNAJC5 in the regulation of α-syn secretion.

## Results

### Reconstitution of DNAJC5-induced α-syn secretion

Previous studies have shown that α-syn secretion can be stimulated by overexpressing DNAJC5 (*Fontaine et al., 2016*). The CS domain of DNAJC5 plays a role in promoting stable membrane attachment based on its overall hydrophobicity and by enabling post-translational palmitoylation catalyzed by membrane-bound Asp-His-His-Cys (DHHC) family palmitoyltransferases (*Greaves and Chamberlain, 2006*). Using a common human cell line, HEK293T, we first tested the expression and subcellular localization of DNAJC5 (*Figure 1B*). Further subcellular enrichment of DNAJC5 was characterized using a C-terminal FLAG tag. Coomassie blue staining revealed two bands of low and high mobility on SDS-PAGE in the membrane and cytosolic fractions, respectively (*Figure 1C*). Similar migration profiles of pamitoylated (P-) and non-palmitoylated (NP-) DNAJC5 have been reported (*Greaves et al., 2012*). We also transfected and fractionated DNAJC5 in other common cell lines including MDA-MB-231 and Hela cells. Compared to DNAJC5 in HEK293T cells, DNAJC5 in MDA-MB-231 and Hela cells appeared predominantly to be in a palmitoylated and membrane-associated form (*Figure 1—figure supplement 1A*). The mobility of P-DNAJC5 in the membrane fraction shifted to that of NP-DNAJC5 after an overnight depalmitoylation reaction with hydroxylamine (HA) (*Figure 1—figure supplement 1B*). Thus, we confirm that membrane anchoring of DNAJC5 requires palmitoylation in our assay.

We next coexpressed DNAJC5 together with α-syn to examine their secretion over time. At 16 hour (h) after transfection, we detected similar basal-levels secretion of α-syn in both DNAJC5-negative and -positive conditions. Two bands of DNAJC5 corresponding to P-DNAJC5 and NP-DNAJC5 were seen in the lysate, but only the P-DNAJC5 was secreted into the medium (*Figure 1D*). The stimulation

of α-syn secretion by DNAJC5 became obvious at a longer incubation time (36 h), and at this time point NP-DNAJC5 was also enriched in the medium (*Figure 1E*). The release of α-syn was not caused by cell death as little to undetected levels of cytoplasmic tubulin found in the culture medium fraction (*Figure 1D and E*). Cell viability was not affected by transfection of different constructs, as shown by trypan blue staining (*Figure 1—figure supplement 2A*). In addition to wild-type (WT) α-syn, secretion of several PD-causing α-syn mutant proteins (A30P, E46K, and A53T) was also induced to differing levels by expression of DNAJC5 (*Figure 1—figure supplement 2B-D*).

In addition to the stimulated secretion of α-syn produced by the expression of exogenous DNAJC5, we examined the dependence of a basal secretion of α-syn on endogenous DNAJC5. We fused α-syn with an N-terminal nanoluciferase (Nluc) (*England et al., 2016*) for sensitive, quantitative detection (*Figure 1—figure supplement 3A*). Stimulated secretion of Nluc-α-syn by overexpression of DNAJC5 was confirmed by immunoblot, indicating that Nluc-fusion did not impede α-syn secretion (*Figure 1—figure supplement 3B*). Without overexpression of DNAJC5, we observed accumulation of Nluc-α-syn signal in the medium over time (*Figure 1—figure supplement 3C*).

Quercetin is a plant-derived flavonoid that has previously been shown to inhibit DNAJC5-mediated trafficking of a bacterial toxin (*Figure 1—figure supplement 3D*; *Deruelle et al., 2021*). We found that quercetin also inhibited Nluc-α-syn secretion in a dose-dependent manner (*Figure 1—figure supplement 3E*), implying a role for endogenous DNAJC5 in α-syn secretion. To exclude the off-target effect of quercetin, we created a DNAJC5 CRISPR knockout (KO) cell line (*Figure 1—figure supplement 3F*). Balfilomycin A1 (BaFA1), a lysosomal ATPase inhibitor, has been shown to stimulate α-syn secretion (*Figure 1—figure supplement 3G*; *Buratta et al., 2020*; *Fernandes et al., 2016*). BaFA1 is also known to stimulate the fusion of lysosomes and multivesicular bodies at the cell surface with the secretion of lysosomal content and exosomes (*Cashikar and Hanson, 2019*; *Hikita et al., 2018*; *Tapper and Sundler, 1995*). BaFA1-stimulated α-syn secretion was confirmed in WT HEK293T cells but substantially reduced in DNAJC5 KO cells (*Figure 1—figure supplement 3H*). Our results suggest that DNAJC5 is required for the secretion of α-syn from the endosome/lysosome and is enhanced by overexpression of DNAJC5.

## Characterization of extracellular DNAJC5 and α-syn

In our established assay, DNAJC5 and α-syn co-secrete into the medium (*Figure 1E*). Secreted α-syn has been reported to be encapsulated inside EVs (*Danzer et al., 2012*). We sought to assess the EV association of secreted α-syn using a medium fractionation protocol based on EV preparations developed by our lab (*Figure 2A*; *Shurtleff et al., 2016*). After serial differential centrifugation, α-syn and NP-DNAJC5 remained soluble. In comparison, P-DNAJC5 co-sedimented with other EV markers after 100k×*g* centrifugation (*Figure 2B*). In addition to WT α-syn, we conducted medium fractionation with several PD-causing α-syn mutants and found they all remained soluble in culture supernatant fractions (*Figure 2—figure supplement 1A*). To test the EV association of P-DNAJC5, we performed a further sucrose step gradient flotation (*Figure 2—figure supplement 2A*) with the 100k high-speed pellet fraction (*Figure 2—figure supplement 2B*). P-DNAJC5 equilibrated with other EV markers to the 10%/40% interface expected for buoyant EVs (*Figure 2—figure supplement 2B*). We conclude that the secreted α-syn induced by DNAJC5 is neither membrane bound nor in a sedimentable fibrillar form.

Next, we sought to characterize the conformation of secreted soluble α-syn. The medium-containing secreted α-syn was pooled, concentrated, and applied to a gel filtration column. Extracellular α-syn eluted from the column at around 60% of the column volume (*Figure 2C*), similar to the elution volume of purified monomeric α-syn (*Figure 2D* and *Figure 2—figure supplement 3E, F*). By an orthogonal assay, we examined the interaction between tagged and untagged forms of secreted α-syn as an indicator of oligomerization. In this assay, equal amounts of plasmids expressing FLAG-tagged α-syn and non-tagged α-syn were co-transfected in HEK293T cells and the medium was collected and incubated with anti-FLAG M2 beads to detect co-immunoprecipitation of the two forms (IP) (*Figure 2E*). FLAG-tagged α-syn migrated more slowly than non-tagged α-syn as detected in samples of the culture medium (*Figure 2F*). We found that only the FLAG-tagged α-syn was immuno-precipitated (*Figure 2F*), suggesting no stable interaction between the two species. The gel filtration and IP assays reinforced our conclusion that α-syn is secreted as a monomer.

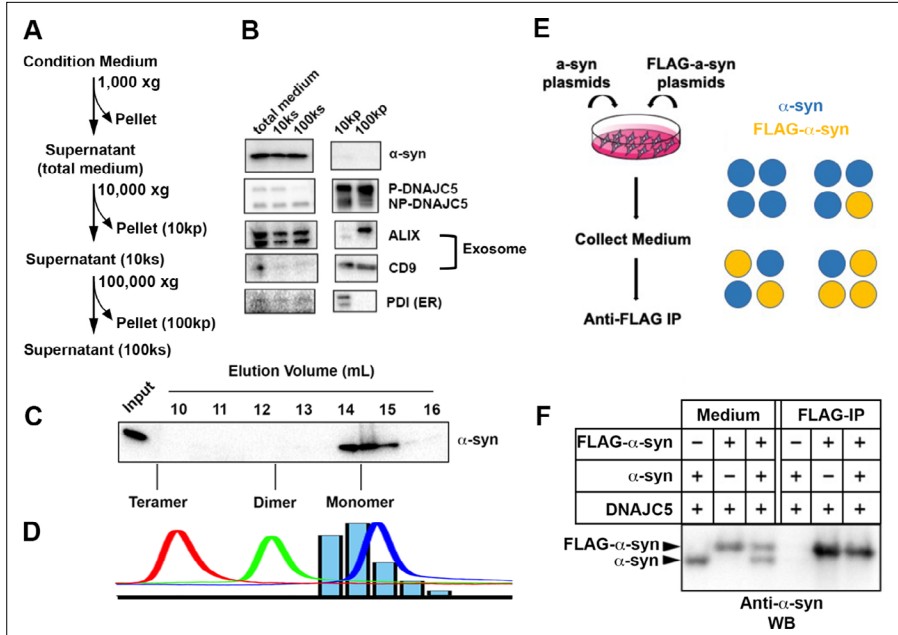

**Figure 2.** Characterization of secreted α-syn. (**A**) Medium fractionation scheme. (**B**) Secreted α-syn was soluble. Differential centrifugation was performed with conditioned medium from HEK293T cells transfected with DNAJC5 and α-syn. Alix and CD9, exosome markers. PDI, an endoplasmic reticulum (ER) marker, was used as exosome-negative control. (**C**) Gel filtration fractionation of medium. Conditioned medium was concentrated and subjected to gel filtration fractionation. Fractions were evaluated by anti-α-syn immunoblot. (**D**) Chromatograms of tandem α-syn monomer (blue curve), dimer (green curve), and tetramer (red curve) were overlaid. In comparison, the relative intensity of secreted α-syn in each fraction was plotted as blue bars. (**E**) Schematic diagram of co-immunoprecipitation (co-IP) of secreted α-syn and FLAG-α-syn. Shown here is possible interaction between α-syn (blue circle) and FLAG-α-syn (yellow circle) in a representative tetrameric conformation. (**F**) Anti-FLAG immunoprecipitation (FLAG-IP) of media from cells transfected with indicated plasmids. Both the medium input and FLAG-IP samples were evaluated with anti-α-syn immunoblot (anti-α-syn WB).

The online version of this article includes the following source data and figure supplement(s) for figure 2:

**Source data 1.** Uncropped immunoblot corresponding to *Figure 2*.

**Figure supplement 1.** Solubility of secreted α-syn variants.

**Figure supplement 1—source data 1.** Uncropped immunoblot images corresponding to *Figure 2—figure supplement 1*.

**Figure supplement 2.** DNAJC5 enriched in buoyant EV fraction.

**Figure supplement 2—source data 1.** Uncropped immunoblot images corresponding to *Figure 2—figure supplement 2*.

**Figure supplement 3.** Assessment of tandem α-syn oligomers by gel filtration chromatography.

**Figure supplement 3—source data 1.** Uncropped gel images corresponding to *Figure 2—figure supplement 3*.

## Secretion of α-syn requires palmitoylation of DNAJC5

Membrane targeting of DNAJC5 is dependent upon palmitoylation (*Greaves et al., 2008*). Two specific mutations, L115R and L116Δ in the CS domain, cause adult-onset neuronal ceroid lipofuscinosis (NCL), a type of neurodegenerative disorder (*Benitez et al., 2011*). NCL mutations reduce the level of DNAJC5 palmitoylation and promote aggregation of the protein (*Diez-Ardanuy et al., 2017*). We perturbed DNAJC5 palmitoylation by either introducing the palmitoylation-deficient mutation L115R or treating cells with the competitive palmitoyl transferase inhibitor, 2-bromopalmitic acid (2-BA) (*Resh, 2006*), and subsequently examined the influence on DNAJC5 membrane association. DNAJC5 palmitoylation largely decreased in the L115R mutant or upon 2-BA treatment. Correspondingly, NP-DNAJC5 accumulated in the cytosol (*Figure 3A and B*).

Having confirmed palmitoylation inhibition by 2-BA and the L115R mutation, we next examined their effect on α-syn secretion. Upon 10 μM 2-BA treatment, α-syn and DNAJC5 secretion were

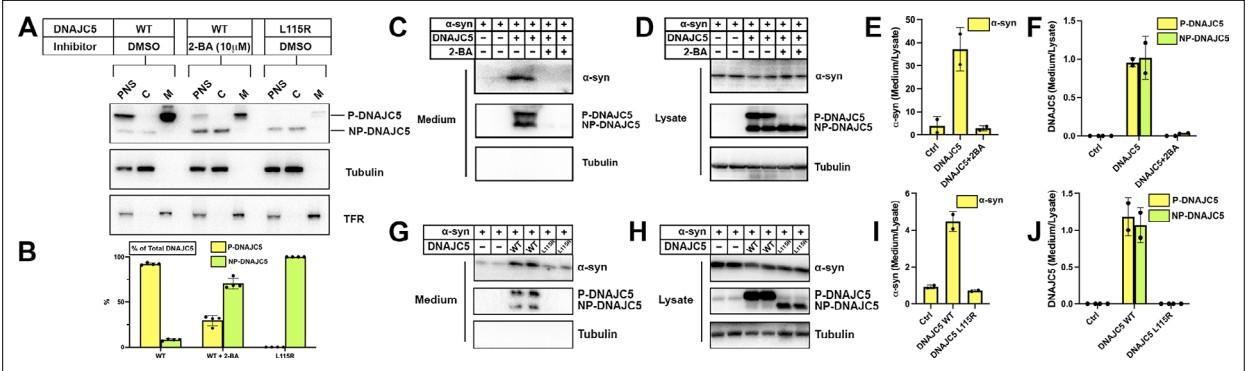

**Figure 3.** Disruption of palmitoylation of DNAJC5 inhibited α-syn secretion. (**A**) Inhibition of DNAJC5 palmitoylation by 2-bromopalmitic acid (2-BA) or introduced mutation L115R. Cellular fractionation was performed with HEK293T cells transfected with WT DNAJC5 and treated with 10 μm 2-BA, or transfected with DNAJC5 L115R mutant. C, cytosol; M, membrane; PNS, post-nuclear supernatant; TFR, transferrin receptor. (**B**) Quantification of the percentage of P-DNAJC5 and NP-DNAJC5 in different conditions as shown in (**A**). Error bars represent standard deviations of three experiments. (**C**) α-syn secretion was blocked with 2-BA treatment. HEK293T cells transfected with indicated plasmids were treated with DMSO or 10 μm 2-BA. Media fractions were collected and secretion was evaluated by SDS-PAGE and immunoblot. (**D**) Palmitoylation of DNAJC5 was blocked in HEK293T cells treated with 2-BA. (**E**) Quantification of normalized α-syn secretion in HEK293T cells after 2-BA treatment. The quantification was based on immunoblot in (**C**) and (**D**). The α-syn secretion was calculated as the amount of α-syn in media divided by the amount in lysate. (**F**) Quantification of normalized DNAJC5 secretion in HEK293T cells after 2-BA treatment. The quantification was based on immunoblot in (**C**) and (**D**). The DNAJC5 secretion was calculated as the amount of DNAJC5 in media divided by the amount in lysate. (**G**) DNAJC5 L115R mutant reduced α-syn secretion compared with WT DNAJC5. Secretion assay with HEK293T cells transfected with indicated plasmids encoding DNAJC5 variant was performed similar to (**C**). (**H**) DNAJC5 L115R was non-palmitoylated in HEK293T cells. (**I**) Quantification of normalized α-syn secretion in HEK293T cells transfected with DNAJC5 L115R mutant. The quantification was based on immunoblot in (**G**) and (**H**). (**J**) Quantification of normalized DNAJC5 secretion in HEK293T cells transfected with DNAJC5 L115R mutant. The quantification was based on immunoblot in (**G**) and (**H**).

The online version of this article includes the following source data and figure supplement(s) for figure 3:

**Source data 1.** Uncropped immunoblot corresponding to *Figure 3*.

**Figure supplement 1.** Dose-dependent inhibition of α-syn secretion by 2-bromopalmitic acid (2-BA).

**Figure supplement 1—source data 1.** Uncropped immunoblot corresponding to *Figure 3—figure supplement 1*.

**Figure supplement 2.** Blockage of USP19-induced α-syn secretion by DNAJC5 L115R or L116Δ mutant.

**Figure supplement 2—source data 1.** Uncropped immunoblot corresponding to *Figure 3—figure supplement 2*.

**Figure supplement 3.** Golgi retention of DNAJC5 in secretion-deficient conditions.

abolished (*Figure 3C, E, F*). The efficacy of the inhibitor was validated by the disappearance of the low-mobility band corresponding to P-DNAJC5 in the cell lysate (*Figure 3D*). Furthermore, 2-BA inhibited DNAJC5 palmitoylation (*Figure 3—figure supplement 1C, E*) and α-syn secretion in a concentration-dependent manner (*Figure 3—figure supplement 1B, D*). Likewise, α-syn secretion was reduced by the palmitoylation-deficient DNAJC5 (L115R) compared with DNAJC5 (WT) (*Figure 3G–J*). DNAJC5 has been proposed to function downstream of the deubiquitinase USP19 in the MAPS pathway (*Xu et al., 2018*). In agreement with the model, DNAJC5 carrying either NCL mutation L115R or L116Δ had no palmitoylated protein band detected in the lysate (*Figure 3—figure supplement 2B*) and blocked USP19-stimulated α-syn secretion (*Figure 3—figure supplement 2A, C*). These results establish that palmitoylation is essential for DNAJC5 membrane association and function in stimulating α-syn secretion.

A previous study reported an altered distribution of DNAJC5 mutant protein to the Golgi apparatus and cytosol (*Nosková et al., 2011*). In confocal immunofluorescence (IF) images, we confirmed that WT DNAJC5 did not colocalize with the Golgi marker GM130, whereas both the L115R mutant and 2-BA treated WT cells partially retained DNAJC5 in puncta coincident with the Golgi marker GM130 (*Figure 3—figure supplement 3A, B, C*). We conclude that α-syn secretion depends upon an appropriate subcellular organelle localization of DNAJC5.

## DNAJC5-dependent internalization of α-syn into enlarged endosomes

DNAJC5 has been reported under normal conditions to be associated with late endosomes (*Lee et al., 2018*). Using confocal IF, we found colocalization between endogenous DNAJC5 and the

late-endosomal marker CD63 (*Figure 4—figure supplement 1A*). To visualize the topological localization of DNAJC5 and α-syn inside or outside endosomes, we turned to a U2OS cell line expressing a fluorescent protein-fused, constitutively active form of Rab5 (Rab5$^{Q79L}$) (*Bohdanowicz et al., 2012*). As a positive control, CD63 localized to the lumen of enlarged endosomes labeled by mCherry-Rab5$^{Q79L}$ (*Figure 4—figure supplement 1B*). We labeled DNAJC5 with the self-labeling HaloTag for multiple choices of color in live-cell imaging (*Los et al., 2008*). Both diffuse and punctate DNAJC5 localized to the lumen of enlarged endosomes (*Figure 4A*). Unlike WT DNAJC5, the DNAJC5 L115R mutant became disperse in the cytosol, rather than being internalized into enlarged endosomes (*Figure 4—figure supplement 2A*). In contrast, mNeonGreen (mNG)-fused α-syn showed diffuse localization in both the cytosol and nucleus but was completely excluded from enlarged endosomes in L115R mutant cells (*Figure 4B*). Notably, we observed the entry of α-syn into enlarged endosomes containing internalized DNAJC5, implying the translocation of α-syn into the membrane compartment required DNAJC5 (*Figure 4C*). The ratio of α-syn-containing endosomes in cells increased significantly with DNAJC5 overexpression (*Figure 4H*). With no luminal localization inside enlarged endosomes, the DNAJC5 L115R mutants also failed to induce entry of α-syn into the same compartments (*Figure 4—figure supplement 2B*). As an independent test of the localization suggested by the imaging results, we applied cell fractionation to separate membranes of DNAJC5- and α-syn-expressing cells (*Figure 4D*). Both DNAJC5 and α-syn were enriched in a 25k membrane pellet fraction (*Figure 4E*). In a protease protection assay with 25k sedimented membranes, we found that α-syn and DNAJC5 were partially resistant to digestion by proteinase K in the absence but not in the presence of Triton X-100 consistent with the conclusion that about half of the proteins were sequestered within membrane compartments (*Figure 4F and G*). Our visual inspection and quantification results are consistent with a membrane translocation role for DNAJC5 prior to α-syn secretion.

## Size and unfolding are dispensable for α-syn secretion

In our medium fractionation assay, secreted α-syn in the extracellular space was characterized as a soluble monomer (*Figure 2*). We generated a series of tandem repeats of α-syn to mimic its oligomeric states (*Figure 5A*; *Dong et al., 2018*). On SDS-PAGE, α-syn tandem repeats showed a larger apparent size than their predicted molecular weights, possibly caused by their extended conformation as intrinsically disordered proteins (*Figure 2—figure supplement 2*). We first determined that these α-syn tandem repeats could also be secreted upon overexpression of DNAJC5 (*Figure 5B*), indicating that DNAJC5 can accommodate α-syn of different sizes. Fractionation of the growth medium showed that secreted tandem repeats were also soluble (*Figure 5—figure supplement 1*).

In conventional and many unconventional secretion processes, the secreted proteins undergo unfolding prior to translocation through a narrow channel across the hydrophobic membrane barrier (*Rapoport et al., 2017*). Recent progress in protein design has allowed the synthesis of super-folded protein constructs (*Kuhlman and Bradley, 2019*). For example, a three helix-bundle protein (3H) designed in the lowest-energy arrangements displayed extreme thermodynamic stability and remained folded even in non-physiological denaturing conditions (*Huang et al., 2014*). Given that mitochondrial protein import is dependent on protein unfolding (*Neupert, 1997*), we tested the effect of 3H insertion on the import of mitochondrial matrix enzyme ornithine transcarbamylase (OTC) (*Horwich et al., 1985*; *Yano et al., 1997*). As a control, we created a pOTC leader peptide fused to GFP (*Figure 5—figure supplement 2A*). pOTC-GFP was enriched in a mitochondria-containing particulate (P) fraction compared to non-tagged GFP. However, a construct in which 3H was inserted between the leader sequence and GFP resulted in 80% of the fusion protein retained in the soluble fraction (*Figure 5—figure supplement 2B, C*). We then used proteinase K protection to assess the topology of pOTC-3H-GFP associated with crude mitochondria (*Figure 5—figure supplement 2D*). About 80% of citrate synthase (CS), a known mitochondria matrix protein, was protected from proteinase K. In contrast, neither the mitochondrial outer membrane protein Tom20 nor pOTC-3H-GFP was protected, suggesting 3H prevented the translocation of GFP into mitochondria (*Figure 5—figure supplement 2E, F*). Using a similar approach, we fused 3H to either the N- or C-terminus of α-syn to impede the unfolding process (*Figure 5D*). These α-syn fusion proteins were expressed and secreted normally into the growth medium (*Figure 5E&F*). These data suggest that protein unfolding is dispensable for α-syn secretion.

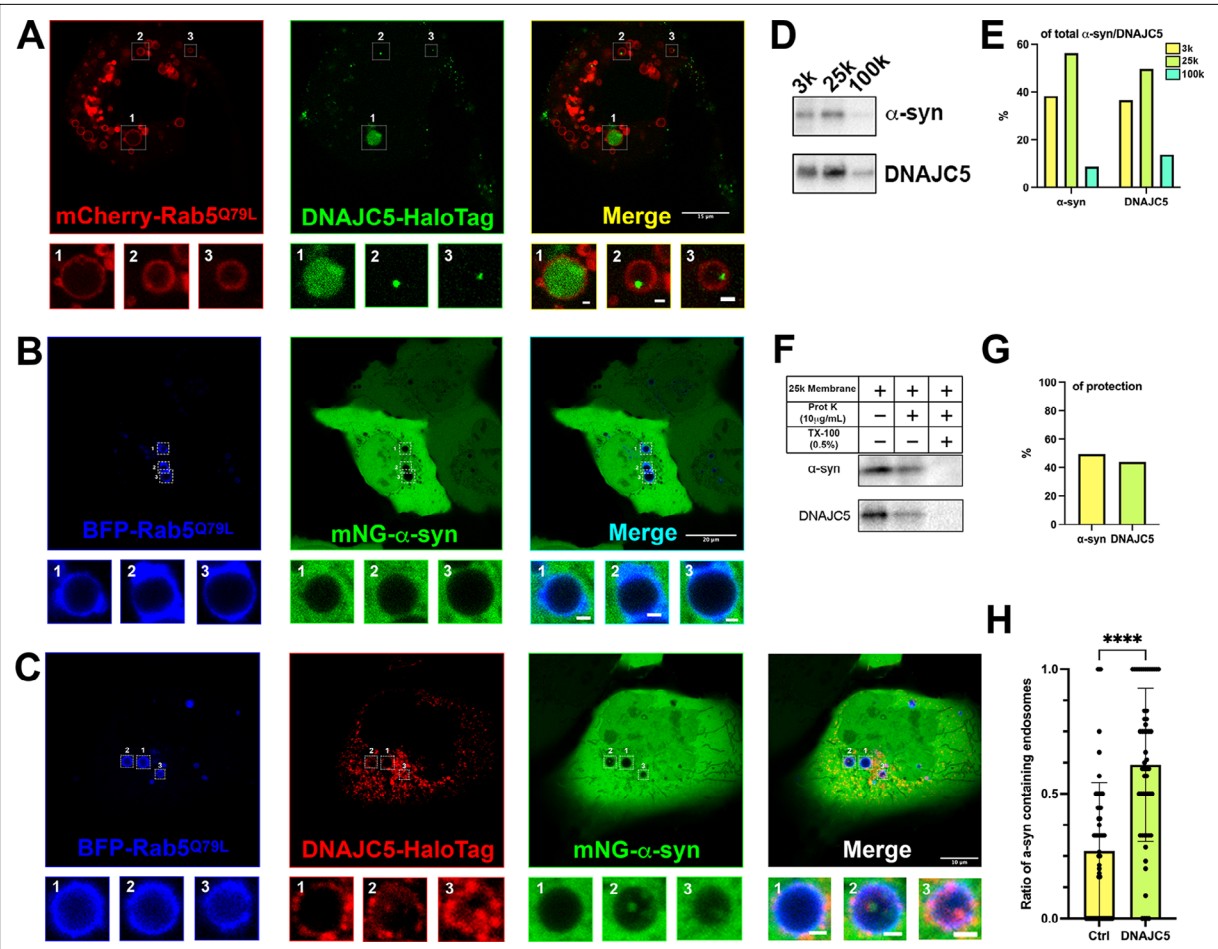

**Figure 4.** Topological localization of α-syn and DNAJC5 in enlarged endosomes. (**A**) DNAJC5 was internalized inside enlarged endosomes. Live U2OS cells expressing mCherry-Rab5$^{Q79L}$ (red) showed circular enlarged endosomes labeled by Rab5 mutant. DNAJC5-HaloTag (green) was visualized by addition of HaloTag Oregon Green Ligand. Representative enlarged endosomes show diffuse (1) or punctate (2 and 3) internalized DNAJC5. Scale bar: 15 μm in overviews and 1 μm in magnified insets. (**B**) α-syn was excluded from enlarged endosomes. In live U2OS cells, expression of BFP-Rab5$^{Q79L}$ (blue) produced enlarged endosomes of similar morphology compared with mCherry-Rab5$^{Q79L}$. mNeonGreen-α-syn (mNG-α-syn, green) was expressed both in the nucleus and cytosol. No mNG-α-syn was found inside enlarged endosomes (1–3). Scale bar: 20 μm in overviews and 1 μm in magnified insets. (**C**) α-syn enters into enlarged endosomes in the presence of DNAJC5. DNAJC5-HaloTag (red) and mNG-α-syn (green) were coexpressed in U2OS cells carrying BFP-Rab5$^{Q79L}$ (blue) mutant and imaged. No mNG-α-syn was internalized in endosome without DNAJC5-HaloTag inside (1). In contrast, mNG-α-syn was found inside endosomes with DNAJC5-HaloTag inside (2 and 3). Scale bar: 10 μm in overviews and 1 μm in magnified insets. (**D**) α-syn and DNAJC5 co-sedimented in membrane fractionation. HEK293T cell homogenate was sequentially centrifuged at increasing velocity from 3000×*g* (3k), 25,000×*g* (25k), and 100,000×*g* (100k). The 25k membrane fraction had the highest amount of both α-syn and DNAJC5. (**E**) Quantification of the membrane fractionation results in (**D**). (**F**) Proteinase K protection assay of 25k membrane-containing α-syn and DNAJC5. (**G**) Quantification of the proteinase K protection assay in (**F**). (**H**) Quantification of the ratio of α-syn-containing endosomes in control cells (no-DNAJC5 transfection) or cells co-transfected with DNAJC5. More than 100 enlarged endosomes were counted in each group. Error bars represent standard deviations. P value<0.0001, two-tailed t test.

The online version of this article includes the following video, source data, and figure supplement(s) for figure 4:

**Source data 1.** Uncropped immunoblot corresponding to *Figure 4*.

**Figure supplement 1.** Immunofluorescence (IF) images of endogenous DNAJC5 and enlarged endosomes.

**Figure supplement 2.** Live-cell images of U2OS cells expressing DNAJC5 L115R mutant and α-syn.

**Figure 4—video 1.** Time lapse of movement of internalized α-syn and DNAJC5 in the enlarged endosomes.
https://elifesciences.org/articles/85837/figures#fig4video1

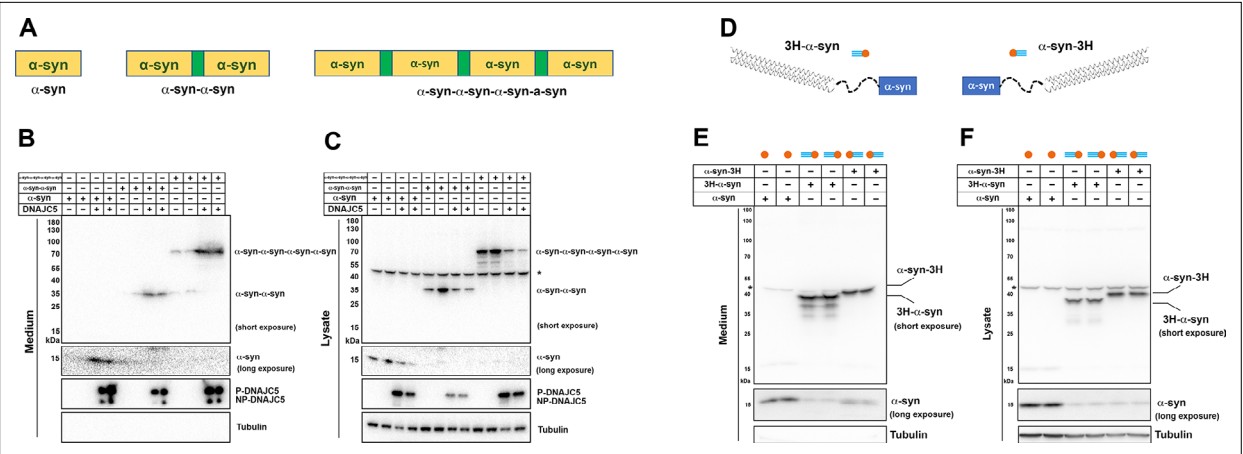

**Figure 5.** Secretion of tandem α-syn oligomers and α-syn fused with thermostable helix-bundle protein. (**A**) Schematic diagrams of tandem α-syn oligomers. α-syn protomers (yellow) were linked head to tail by flexible linker (green) to mimic increased size of α-syn oligomers. (**B**) Secretion of tandem α-syn oligomers in medium. Secretion assay was performed with media from HEK293T cells transfected with indicated tandem α-syn oligomers. Tandem α-syn oligomers are more sensitively detected by immunoblot which were exposed for shorter time compared with WT α-syn. (**C**) Expression of tandem α-syn oligomers in HEK293T cells. *a non-specific band. (**D**) Schematic diagrams of N-terminal fused and C-terminal fused thermostable three helix-bundle (3H-) α-syn. 3H shown as three blue dashes, α-syn shown as orange circle. (**E**) Secretion of 3H-α-syn and α-syn-3H in medium. Secretion assay was performed with media from HEK293T cells transfected with indicated 3H-fused α-syn constructs. *a non-specific band. (**F**) Expression of 3H-α-syn and α-syn-3H in HEK293T cells. *a non-specific band.

The online version of this article includes the following source data and figure supplement(s) for figure 5:

**Source data 1.** Uncropped immunoblot corresponding to *Figure 5*.

**Figure supplement 1.** Medium fractionation of secreted tandem α-syn oligomers.

**Figure supplement 1—source data 1.** Uncropped immunoblot corresponding to *Figure 5—figure supplement 1*.

**Figure supplement 2.** Thermostable three helix bundle blocked pOTC-mediated mitochondrial import of GFP.

**Figure supplement 2—source data 1.** Uncropped immunoblot corresponding to *Figure 5—figure supplement 2*.

## XPACK fusion rescues DNAJC5 L115R secretion deficiency by induced oligomerization

DNAJC5 has been reported to have an intrinsic propensity to form SDS-resistant oligomers (*Zhang and Chandra, 2014*). In a whole gel immunoblot of extracellular DNAJC5, we noticed many diffuse, ladder-like bands that migrated more slowly than the two corresponding to P-DNAJC5 and NP-D-NAJC5 (*Figure 6A*), possibly higher molecular weight (HMW) oligomers. This apparent oligomerization of DNAJC5 became more obvious when the J domain was deleted (*Figure 6—figure supplement 1A*). The migration of HMW-DNAJC5 was not altered in samples heated in the presence of a reducing agent (*Figure 6—figure supplement 1B*). To assess the size of these HMW species of DNAJC5, we evaluated a cell lysate by gel filtration chromatography. HMW-DNAJC5 fractionated according to its apparent size, forming a stair-like pattern on the DNAJC5 immunoblot (*Figure 6B*). HMW-DNAJC5 chromatographed within the gel filtration column volume, consistent with discrete protein species rather than aggregates (*Figure 6B*). These results suggest the presence of higher-order, SDS-resistant, non-disulfide-bonded DNAJC5 oligomers both in intracellular and extracellular fractions.

XPACK (XP) is a membrane-targeting peptide sequence used widely in studies of cargo loading into exosomes and delivery to target cells of choice (*Yim et al., 2016*). The exosome loading process by XP is also dependent on two lipidation reactions—myristoylation on the first glycine and palmitoylation on the second cysteine (*Figure 6—figure supplement 1C*; *Zacharias et al., 2002*). Given the similarity of membrane localization and lipidation between XP and the CS domain of DNAJC5, we examined α-syn secretion in cells expressing an XP-DNAJC5 fusion. An N-terminal XP fusion (XP-WT) resulted in the expression of a species that migrated at the position of P-DNAJC5, in contrast to the two species representing P- and NP-DNAJC5 in the WT DNAJC5 sample (*Figure 6—figure supplement 1D*). Again in contrast to WT DNAJC5, XP-DNAJC5 was exclusively associated with the sedimentable membrane fraction (*Figure 6—figure supplement 1E*). This suggested that XP-mediated lipidation

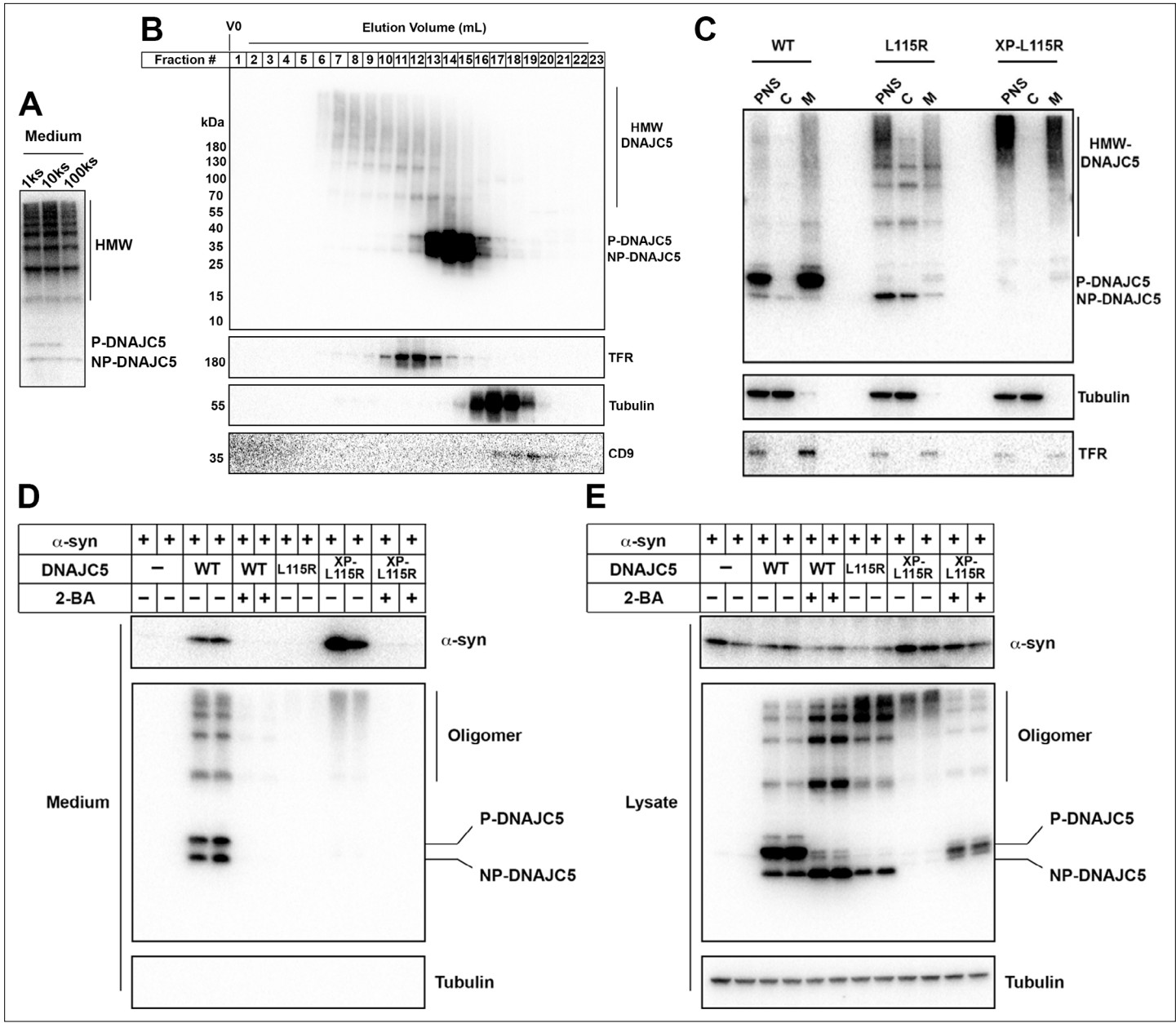

**Figure 6.** XPACK (XP)-induced DNAJC5 L115R oligomerization rescued α-syn secretion. (**A**) Ladder pattern of higher molecular weight (HMW) DNAJC5 oligomers in the medium. Medium from HEK293T cell culture transfected with DNAJC5 was centrifuged at 1000 (1k)×*g*, 10,000 (10k)×*g*, and 100,000 (100k)×*g*, followed by SDS-PAGE and immunoblot of supernatant (s) fractions at each centrifugation step. (**B**) Fractionation of HMW-DNAJC5 with gel filtration. HEK293T cells transfected with DNAJC5 were lysed, clarified, and subjected to gel filtration. HMW-DNAJC5 of different sizes were separated based on their corresponding molecular weight. (**C**) XP-DNAJC5 L115R mutant forms a membrane-bound oligomer. Cellular fractionation was performed with HEK293T cells transfected with indicated DNAJC5 variants. Note the substantial change of electrophoretic mobility of XP-DNAJC5 L115R on SDS-PAGE. (**D**) α-syn secretion induced by XP-DNAJC5 L115R. Secretion assay was performed with HEK293T cells transfected with indicated plasmids. About 10 μm 2-BA was used to block induced α-syn. (**E**) Expression of α-syn and DNAJC5 variants in HEK293T cells. Note the substantial change in electrophoretic mobility of 2-BA-treated XP-DNAJC5 L115R on SDS-PAGE.

The online version of this article includes the following source data and figure supplement(s) for figure 6:

**Source data 1.** Uncropped immunoblot corresponding to *Figure 6*.

**Figure supplement 1.** Characterization of HMW-DNAJC5 and XPACK fusion.

**Figure supplement 1—source data 1.** Uncropped immunoblot corresponding to *Figure 6—figure supplement 1*.

**Figure supplement 2.** Live-cell images of U2OS cells expressing XP-DNAJC5 L115R mutant and α-syn.

was highly efficient and possibly irreversible. Formation of the lower electrophoretic mobility and membrane-associated form of XP-DNAJC5 was blocked by 2-BA treatment, indicating XP lipidation included palmitoylation (*Figure 6—figure supplement 1E*). XP fusion to the palmitoylation-deficient mutant of DNAJC5 (L115R) did not produce a species that migrated at the position of P-DNAJC5 but resulted in several less abundant species that migrated between the positions of NP- and P-DNAJC5 (*Figure 6—figure supplement 1D*). We introduced a serine to leucine point mutation in the XPACK sequence which was predicted to block lipidation (dead XPACK, DXP) (*Figure 6—figure supplement 1C*). The DXP-DNAJC5 L115R species had the same mobility as NP-DNAJC5 (*Figure 6—figure supplement 1D*).

We conducted cellular fractionation on lysates of cells expressing DNAJC5 XP-L115R. XP-L115R was highly enriched in the membrane fraction, likely as a result of XPACK-mediated lipidation (*Figure 6C*). SDS-PAGE of XP-L115R released by detergent solubilization migrated slowly and remained near the top of the gel, suggesting XPACK-induced high-order oligomerization or aggregation (*Figure 6C*). In spite of the apparent difference between DNAJC5 XP-L115R and WT DNAJC5, α-syn secretion was stimulated by the expression of both species (*Figure 6D*). Treatment with the palmitoylation inhibitor 2-BA resulted in the formation of XP-L115R that migrated to a position similar to that of monomeric DNAJC5 (*Figure 6E*). Correspondingly, secretion of α-syn was no longer stimulated by the palmitoylation deficient DNAJC5 XP-L115R monomer (*Figure 6D*).

In order to expand on the fractionation results, we employed confocal microscopy to examine the subcellular localization of DNAJC5 XP-L115R fused with a C-terminal HaloTag. In contrast to the diffuse distribution of the DNAJC5 L115R mutant, which was excluded from the interior of enlarged endosomes (*Figure 4—figure supplement 2A*), punctate DNAJC5 XP-L115R was widely associated with enlarged endosomes (*Figure 6—figure supplement 2A*). Internalization events were found in several enlarged endosomes (*Figure 6—figure supplement 2A*, magnified insets). mNG-α-syn was also incorporated into endosomal compartments in cells coexpressing DNAJC5 XP-L115R (*Figure 6—figure supplement 2B*). The level of α-syn-containing endosomes in cells expressing DNAJC5 XP-L115R was ~2-fold higher than in cells expressing the DNAJC5 L115R mutant (*Figure 6—figure supplement 2C*). Our imaging data corroborate the biochemical similarity between WT DNAJC5 and DNAJC5 XP-L115R.

## Secretion of endogenous α-syn from neurons is mediated by DNAJC5

To evaluate the function of DNAJC5 in α-syn secretion at physiological levels of expression in a neuronal cell line, we employed SH-SY5Y, a neuroblastoma line that differentiates in the presence of retinoic acid (RA) into nerve cells that express dopamine neuron (DA) markers including tyrosine hydroxylase (TH) (*Lopes et al., 2010*). We observed elevated levels of expression of α-syn and dopamine transporter (DAT) in SH-SY5Y cells after 6 days of RA-induced differentiation (*Figure 7—figure supplement 1A*). Fractionation of SH-SY5Y cell lysates resolved DNAJC5 into the low electrophoretic mobility P form associated with sedimentable membranes (M) and the non-sedimentable cytosolic NP form (*Figure 7A*). Hydroxylamine treatment of the membrane-associated form converted DNAJC5 to the electrophoretic mobility position of the NP form, as before (*Figure 7—figure supplement 1B*; *Figure 1—figure supplement 1B*).

We employed a sensitive α-syn enzyme-linked immunosorbent assay (ELISA) and detected about 300 pg/ml α-syn secreted into the supernatant of RA-differentiated SH-SY5Y cells (*Førland et al., 2018*; *Figure 7B*). Differential centrifugation of the medium fraction demonstrated that the bulk of the secreted α-syn remained soluble (*Figure 7B*). A 100k pellet fraction was probed by protease protection for the localization of Flot-2, an exosome marker and DNAJC5. Both were resistant to degradation by proteinase K in the absence but not in the presence of TX-100, suggesting that both were protected within the lumen of EVs (*Figure 7—figure supplement 1C*). In contrast, residual full-length (FL) α-syn in the pellet fraction was cleaved by proteinase K without or with detergent (*Figure 7—figure supplement 1C*). As an additional test, sucrose gradient flotation of the high-speed pellet fraction as used in *Figure 2—figure supplement 2A* revealed that α-syn secreted by differentiated SH-SY5Y cells was not buoyant whereas DNAJC5 appeared associated with membranes fractionating at the position of EVs (*Figure 7—figure supplement 1D*). Thus, as with HEK293T cells, α-syn secreted by differentiated SH-SY5Y cells appears not to be enclosed within EVs.

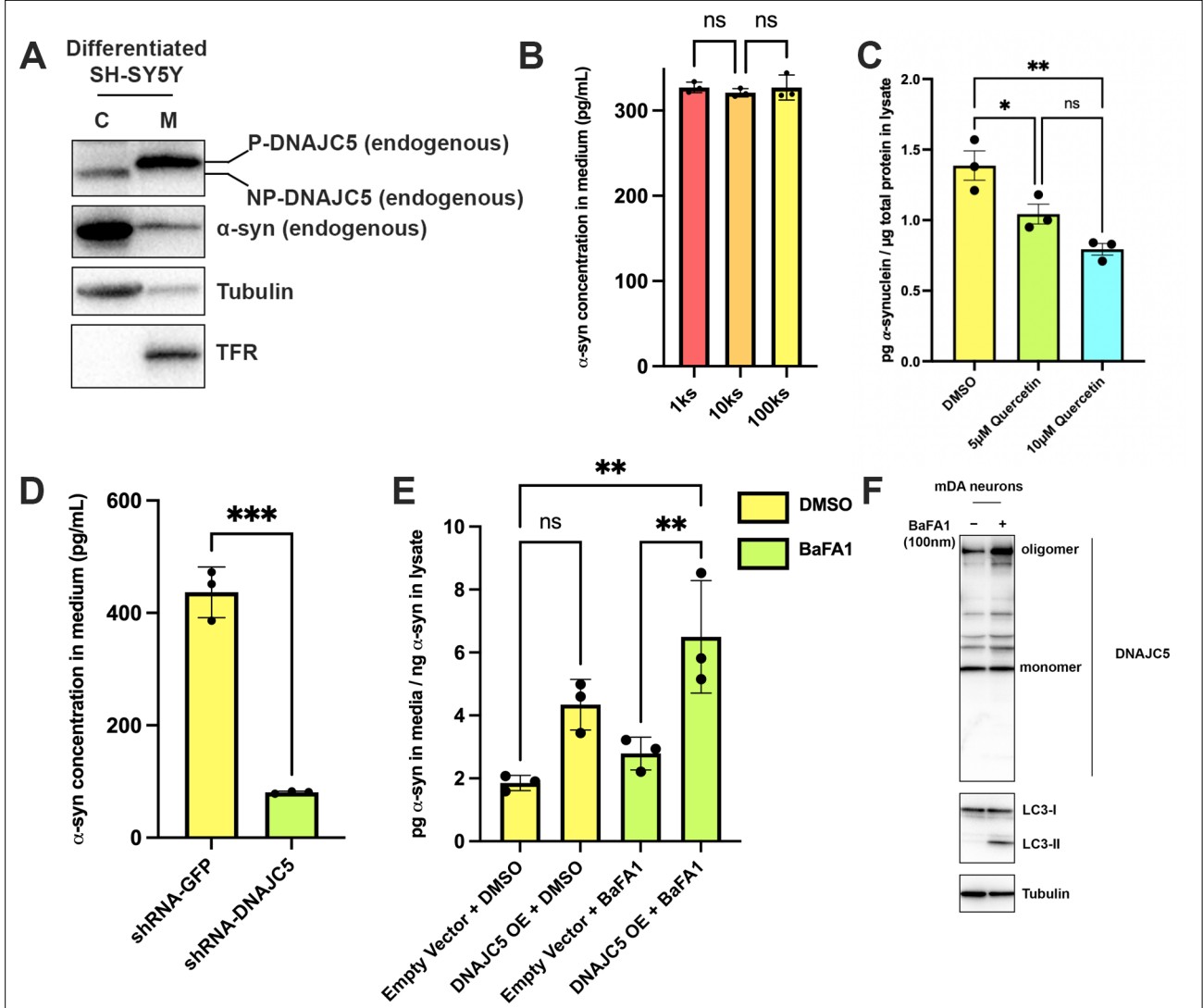

**Figure 7.** Recapitulation of endogenous DNAJC5-mediated α-syn secretion in various neuronal cell cultures. (**A**) Membrane and cytosol fractionation of differentiated SH-SY5Y neuroblastoma cells. The fractionation was performed as depicted in *Figure 1B*. C, cytosol; M, membrane. The distribution of endogenous DNAJC5 and α-syn was evaluated by immunoblot. Transferrin receptor (TFR) was used as a membrane marker. Tubulin was used as a cytosol marker. (**B**) Quantification of α-syn level in the supernatant of centrifuged media with ELISA. Conditioned media were collected and sequentially centrifuged at 1000 (1k)×*g*, 10,000 (10k)×*g*, and 100,000 (100k)×*g*. The supernatant from each centrifugation step (1ks, 10ks, and 100ks) was collected and measured by LEGEND MAX Human α-synuclein (Colorimetric) ELISA Kit. One-way ANOVA showed no significant (ns) difference of α-syn level between fractions. (**C**) Quercetin inhibited endogenous α-syn secretion in hiPSC-derived midbrain dopamine neurons. hiPSC-dopamine neurons carrying the *GBA-N370S* mutation were treated with quercetin (5 μM or 10 μM) at day 35. Culture media samples were harvested after 3 days treatment at day 38 and α-syn levels in the media were analyzed by electro-chemiluminescent immunoassay. Data points represent individual cell lines derived from different donors and are normalised to total protein in the corresponding cell lysates. One-way ANOVA followed by Tukey's post hoc test shows a significant reduction in α-syn secretion with increasing quercetin concentration (*p<0.05, **p<0.01). (**D**) Depletion of endogenous DNAJC5 in SH-SY5Y cells decreased basal α-syn secretion. After 3 days of culture, the media from differentiated SH-SY5Y cells transduced with shRNA targeting GFP (shRNA-GFP) or shRNA targeting DNAJC5 (shRNA-DNAJC5) were collected and the extracellular α-syn was quantified with ELISA. P value<0.0002, two-tailed t test. (**E**) Expression of exogenous human DNAJC5 in mouse mDA stimulated basal α-syn secretion. WT mDA and mDA expressing hDNAJC5 were treated with DMSO or 100 nM BaFA1. Quantification of α-syn in conditioned media was performed with Mouse α-synuclein ELISA Kit (Abcam). α-syn secretion was normalized by dividing the α-syn in media (pg/ml) by the α-syn in cell lysates (ng/ml). P value<0.01, one-way ANOVA. (**F**) BaFA1 increased DNAJC5 oligomerization in mouse mDA neurons.

The online version of this article includes the following source data and figure supplement(s) for figure 7:

**Source data 1.** Uncropped immunoblot corresponding to *Figure 7*.

**Figure supplement 1.** Basal α-syn secreted as a soluble form from differentiated SH-SY5Y.

*Figure 7 continued on next page*

*Figure 7 continued*

**Figure supplement 1—source data 1.** Uncropped immunoblot corresponding to *Figure 7—figure supplement 1*.

**Figure supplement 2.** Differentiation of human induced pluripotent stem cells (hiPSCs) and palmitoylation of DNAJC5.

**Figure supplement 2—source data 1.** Uncropped immunoblot corresponding to *Figure 7—figure supplement 2*.

**Figure supplement 3.** shRNA-mediated DNAJC5 knockdown in differentiated SH-SY5Y cells.

**Figure supplement 3—source data 1.** Uncropped immunoblot corresponding to *Figure 7—figure supplement 3*.

**Figure supplement 4.** Differentiation of mouse embryonic stem cells (mESCs) and expression of human DNAJC5.

**Figure supplement 4—source data 1.** Uncropped immunoblot corresponding to *Figure 7—figure supplement 4*.

Midbrain dopamine (mDA) neurons (*Figure 7—figure supplement 2A*) differentiated from human-induced pluripotent stem cells (hiPSCs) from Parkinson's patients with the *GBA-N370S* mutation, a genetic lesion that causes ER stress and dysfunctional lysosomes, release about twice the level of α-syn compared to control WT neurons (*Fernandes et al., 2016*; *Lang et al., 2019*). Treatment of $GBA^{N370S}$ hiPSC-derived dopamine neurons with the DNAJC5 inhibitor quercetin led to a significant dose-dependent reduction in the secretion of endogenous α-syn (*Figure 7C*). Immunoblotting of hiPSC-derived dopamine neuron lysate revealed that the endogenous DNAJC5 is natively palmitoylated which can be partially reduced by treatment with 2-BA (10 µM) to induce the formation of the lower, non-palmitoylated band (*Figure 7—figure supplement 2B*). However, this partial de-palmitoylation of DNAJC5 was insufficient to inhibit α-syn release by iPSC-derived *GBA-N370S* dopamine neurons at the concentration of 2-BA used (*Figure 7—figure supplement 2C*).

To examine the role of DNAJC5 in differentiated SH-SY5Y cells, we silenced the expression of the chromosomal locus by small hairpin RNAs (shRNAs) transduced by lentivirus. The efficiency of shRNA targeting DNAJC5 was confirmed by knockdown (KD) of endogenous DNAJC5 in HEK293T cells (*Figure 7—figure supplement 3A*). Similarly, DNAJC5 was successfully depleted in differentiated SH-SY5Y cells (*Figure 7—figure supplement 3B*). As a result, secretion of α-syn was reduced fivefold compared with a control transduced with shRNA targeting GFP (*Figure 7D*).

In HEK293T cells, overexpression of DNAJC5 increased α-syn secretion (*Figure 1E*). We were unable to observe enhanced secretion of α-syn in SH-SY5Y overexpressing DNAJC5, possibly because of a high level of expression of endogenous DNAJC5 in the differentiated cells (data not shown). To test the effect of DNAJC5 on the basal level of α-syn secretion in neurons, we stably transduced mouse embryonic stem cells (mESCs) with lentivirus-containing human DNAJC5 WT or L115R and differentiated them into mDA neurons (*Figure 7—figure supplement 4A*). After differentiation, DNAJC5 WT was expressed in mDA, but we were unable to detect the expression of DNAJC5 L115R (*Figure 7—figure supplement 4B*). Analysis of conditioned media by ELISA revealed a twofold elevated α-syn secretion in DNAJC5 WT overexpressing mDA compared to control with an empty vector in the presence of BaFA1 (*Figure 7E*). With both BaFA1 treatment and DNAJC5 overexpression, α-syn secretion was increased threefold (*Figure 7E*). We examined the cell lysate of mDA by immunoblot. The effect of BaFA1 inhibition was indicated by the appearance of a lipidated form of LC3 (LC3-II) (*Figure 7F*). BaFA1 treatment also induced more DNAJC5 oligomer formation (*Figure 7F*). We conclude that DNAJC5 stimulates α-syn secretion in differentiated DA neurons as it does in HEK293T cells.

## The J domain and C-terminal tail (C tail) of DNAJC5 are dispensable for α-syn secretion

The secretion deficiency caused by the L115R mutation highlights the importance of the CS domain of DNAJC5 in regulating α-syn secretion. The structure of the J domain of DNAJC5 has been solved by nuclear magnetic resonance (NMR) (*Patel et al., 2016*). Recent progress in deep learning algorithms, exemplified by AlphaFold, enables atomic accuracy in protein structure prediction (*Jumper et al., 2021*). We searched the public AlphaFold database to examine the predicted structure of FL DNAJC5. In the predicted structure, the J domain showed a conserved overall globular J protein fold within the N-terminus, linked by the helical CS domain to the flexible C-terminal tail. Only a short helix was predicted to reside within the C-tail (*Figure 8A*). We refined the boundary of each domain in DNAJC5 based on the predicted structure.

Using this information, we showed that oligomerization of DNAJC5 increased when the J domain was deleted (*Figure 6—figure supplement 1A* and *Figure 8C*). As recently reported by *Lee et al.,*

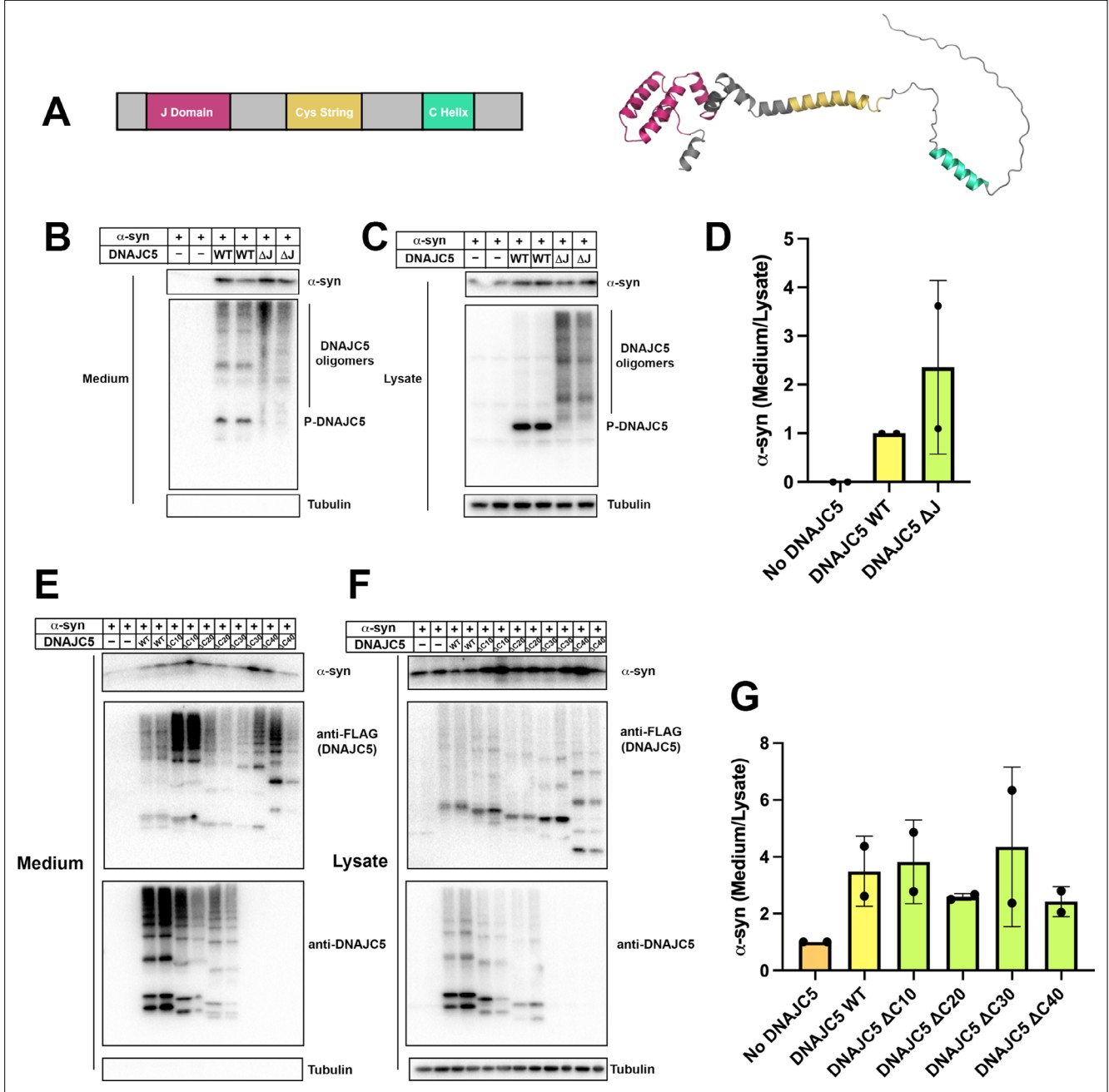

**Figure 8.** Domain mapping of secretion-competent DNAJC5. (**A**) Predicted structure of DNAJC5 by AlphaFold. Color scheme: J domain (magenta), Cys string domain (yellow) and C-terminal helix (green). (**B**) DNAJC5 (ΔJ) was competent to induce α-syn secretion into the medium. HEK293T cells were transfected with indicated plasmids. Media were collected after 36 hr and evaluated with immunoblot. (**C**) DNAJC5 (ΔJ) formed oligomers in HEK293T cells. (**D**) Quantification of normalized α-syn secretion in HEK293T cells transfected with WT DNAJC5 or DNAJC5 (ΔJ). Quantification was based on immunoblot in (**B**) and (**C**). The α-syn secretion was calculated as the amount of α-syn in media divided by the amount in lysate. α-syn secretion in cells transfected with WT DNAJC5 was normalized as 1. (**E**) C-terminal truncated DNAJC5 constructs were competent to induce α-syn secretion in the medium. HEK293T cells were transfected with C-terminal truncated DNAJC5 and α-syn. DNAJC5 antibodies cannot recognize DNAJC5 (ΔC30) and DNAJC5 (ΔC40) because of a missing epitope in the C-terminus. Instead, DNAJC5 (ΔC30) and DNAJC5 (ΔC40) were detected by C-terminal FLAG tags. All the C-terminal truncated DNAJC5 constructs showed smear-like oligomers. (**F**) Expression of C-terminal truncated DNAJC5 constructs in HEK293T cells. Immunoblot of anti-FLAG antibody and anti-DNAJC5 antibody cross-validated the existence of oligomers. (**G**) Quantification of normalized α-syn secretion in HEK293T cells transfected with WT DNAJC5 or different C-terminal truncated DNAJC5 constructs (ΔC10, ΔC20, ΔC30, and ΔC40). Quantification was based on immunoblot in (**E**) and (**F**). The α-syn secretion was calculated as the amount of α-syn in media divided by the amount in lysate. α-syn secretion in cells without DNAJC5 transfection was normalized as 1.

The online version of this article includes the following source data for figure 8:

*Figure 8 continued on next page*

*Figure 8 continued*

**Source data 1.** Uncropped immunoblot corresponding to *Figure 8*.

*2022*, deletion of the J domain increased the level of α-syn secretion induced by DNAJC5 (*Figure 8B and D*). Next, we examined the function of the C-tail by truncating about 10 AAs at a time, resulting in a series of C-terminal truncated DNAJC5 constructs, that is, DNAJC5 ΔC10, ΔC20, ΔC30, and ΔC40. All four DNAJC5 C-terminal truncations were expressed and formed oligomers in cells (*Figure 8F*). C-terminal truncated oligomers were co-secreted with α-syn into the medium (*Figure 8E and G*). This result demonstrates that neither the J domain nor the C-tail is required for DNAJC5 to induce α-syn secretion.

## Discussion

Transmission of protein aggregates and subsequent self-amplification is emerging as a common theme across various neurodegenerative diseases. DNAJC5 has been shown to control the release of neurodegenerative disease proteins but the mechanism of action of this protein in unconventional

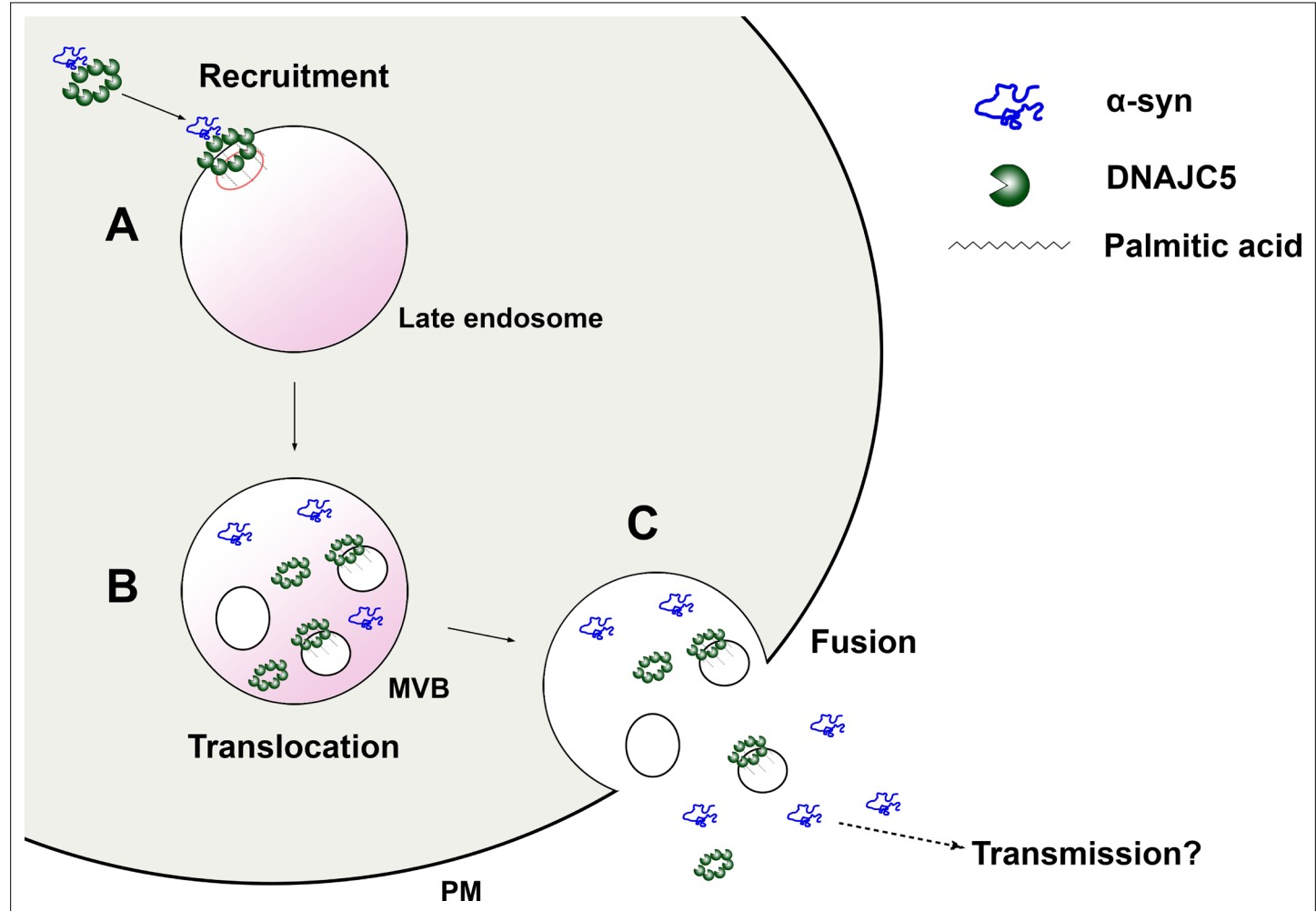

**Figure 9.** A model for palmitoylated DNAJC5 oligomer-mediated α-syn secretion. (**A**) Recruitment of α-syn on the membrane by DNAJC5. DNAJC5 binds to α-syn and targets it to late endosomes by palmitoylation. DNAJC5 forms a high-order oligomer to accommodate α-syn. (**B**) Translocation of α-syn and DNAJC5 into the membrane compartment. Both α-syn and DNAJC5 are translocated into the endosome lumen along with intraluminal vesicles (ILVs), forming a multivesicular body (MVB). (**C**) Secretion of α-syn and DNAJC5. Upon fusion between MVB and plasma membrane (PM), the cargos are expelled into the extracellular space. α-syn is soluble. DNAJC5 exists in both soluble and membrane-bound forms. Further transmission potentially occurs after secretion.

secretion remains elusive. In this study, we reconstituted DNAJC5-regulated α-syn secretion in cultured HEK293T cells, in RA neuronally differentiated human cells and in hiPSC-derived midbrain DA neurons. By combining this assay with medium and cellular fractionation, we demonstrated that membrane-anchoring of DNAJC5 through palmitoylation is crucial for its secretion and the secretion of α-syn as a soluble monomer. In addition, we observed the topological locations of DNAJC5 and α-syn within enlarged endosomes, presumably at an intermediate stage prior to secretion. Furthermore, DNAJC5 was found to form oligomers and the importance of the oligomerization was highlighted by the use of a lipidated XPACK fusion peptide. Our findings on the role of DNAJC5 extend to differentiated DA neurons of human and mouse origin. Finally, we provide evidence that both palmitoylation and oligomerization are solely dependent on the CS domain, which is required for α-syn secretion. Based on our biochemical assays and imaging observations, we propose that palmitoylated DNAJC5 oligomers function at a step involving membrane translocation of cytosolic α-syn, enabling it to become competent for secretion (*Figure 9*).

The in vivo toxicity of α-syn aggregates remains elusive (*Lashuel et al., 2013*), but its propagation accompanies the progression of PD (*Braak et al., 2006*; *Braak et al., 2003*). Recently, *Caló et al., 2021* found that DNAJC5 expression decreases in α-syn transgenic mice. Overexpression of DNAJC5 in vivo is reported to rescue α-syn aggregation-dependent pathology and increase the accumulation of monomeric α-syn (*Caló et al., 2021*). As we find and others have reported, iPSC-derived neurons also secrete α-syn in a largely soluble form (*Fernandes et al., 2016*). Using the criteria of differential sedimentation and gel filtration chromatography, we conclude that α-syn is secreted in cultures cells as a soluble monomeric species not enclosed within EVs, regardless of mutations modeled on PD (*Figure 2—figure supplement 1A*) or as expressed in tandem arrays or in gene fusions to tightly folded proteins (*Figure 5—figure supplement 1*). Consistent with our results, other MAPS substrates are also reported to be secreted in a soluble form (*Lee et al., 2016*). Although α-syn oligomers have also been found in EVs (*Danzer et al., 2012*; *Emmanouilidou et al., 2010*; *Guo et al., 2020*), we see no evidence for this in our culture medium fractionation and immunoblot experiments (*Figure 2B*). With a more sensitive and quantitative Nluc assay, 85% of secreted α-syn was found to be soluble (*Figure 2—figure supplement 1B*). Similarly, the basal level of α-syn secreted by differentiated neuroblastoma cells is mainly soluble and not detected within EVs (*Figure 7—figure supplement 1A*).

The release of soluble α-syn may be an early event in pathogenesis of PD, prior to the deposition of aggregates. Inhibition of the lysosomal ATPase with bafilomycin A (BaFA1) is known to induce lysosome fusion to the cell surface and secretion of lysosomal content including both soluble and aggregate forms of α-syn (*Xie et al., 2022*; *Figure 1—figure supplement 3H*, *Figure 7E*). Such secretion may be part of a cell protective mechanism but it may also promote the interneuronal spread of monomer and oligomer.

Numerous neuronal proteins are palmitoylated, including synaptic scaffolding proteins, signaling proteins, and synaptic vesicle proteins (*Linder and Deschenes, 2007*). Protein palmitoylation has been implicated in the pathogenesis of neurodegenerative diseases (*Cho and Park, 2016*). In PD particularly, a recent study reported that upregulation of cellular palmitoylation decreased α-syn cytoplasmic inclusions (*Ho et al., 2021*). The neuropathology and behavior deficiency of Huntington disease (HD) mice can be reversed by boosting brain palmitoylation (*Virlogeux et al., 2021*). In the case of DNAJC5, L115R, and L116Δ, the two adjacent mutations causing decreased palmitoylation of DNAJC5 monomers, lead to a familial form of NCL (*Benitez et al., 2011*; *Diez-Ardanuy et al., 2017*). Our results suggested that the secretion of neurodegenerative disease proteins is also dependent on palmitoylation, possibly alleviating the cellular burden of protein aggregate accumulation. Notably, the general inhibition of cellular palmitoylation by 2-BA led to a complete block of α-syn secretion, whereas the specific palmitoylation deficient DNAJC5 mutant L115R only partially decreased α-syn secretion (*Figure 3*). The difference implies the existence of palmitoylation-dependent factors other than DNAJC5.

Although DNAJC5 coexpressed with α-syn and palmitoylation are required for secretion, EV-associated P-DNAJC5 clearly separated from soluble α-syn in the culture medium (*Figure 2B* and *Figure 2—figure supplement 2B*). At which step do the two separate? In our live-cell imaging experiments, the internalized DNAJC5 inside enlarged endosomes had both punctate and diffuse distributions (*Figure 4A*). This may represent the soluble NP-DNAJC5 and membrane-attached P-DNAJC5, respectively. In the time-lapse imaging of internalized α-syn induced by DNAJC5, both DNAJC5 and

α-syn moved dynamically inside the compartment, without significant colocalization (*Figure 4—video 1*). This observation suggests that the separation of DNAJC5 and α-syn may occur prior to their secretion when the late endosome and plasma membrane fuse.

*Zhang et al., 2020* have reported a novel membrane channel, TMED10, for the unconventional secretion of IL-1β. These authors speculate an activation-on-demand oligomerization of TMED10 membrane subunits to form a conducting channel for substrate translocation, a process they refer to as THU (*Zhang et al., 2020*). α-syn was reported to not depend on TMED10 for unconventional secretion (*Zhang et al., 2020*). Similarly, in chaperone-mediated autophagy (CMA), cytosolic substrates are proposed to be translocated into lysosomes through a channel formed by the oligomerization of a single-transmembrane protein, LAMP2A (*Bandyopadhyay et al., 2008*). The secretion of α-syn has been shown to be independent of CMA (*Lee et al., 2018*). Without a transmembrane domain, membrane-tethered P-DNAJC5 oligomer is unlikely to be a channel for translocation. A recent report identified CD98hc, an AA transporter subunit, to be a DNAJC5 interactor that is required for α-syn secretion (*Lee et al., 2022*). It remains to be determined whether CD98hc or other as yet uncharacterized membrane proteins are directly involved in α-syn membrane translocation.

In SEC61-mediated co-translational translocation, substrates enter the SEC61 translocon in an unfolded state (*Rapoport et al., 2017*). In THU and CMA, substrate unfolding is also required for translocation across the membrane (*Kaushik and Cuervo, 2018*; *Zhang et al., 2020*). In striking contrast, unfolding and size is not a limiting factor for α-syn secretion (*Figure 5*). As a precedent of translocation without unfolding, studies have shown the import of folded proteins into the matrix of peroxisomes and obviously through the nuclear pore (*Kim and Hettema, 2015*; *Lin and Hoelz, 2019*). DNAJC5 forms a series of extremely stable oligomers, which may provide versatile adaptors to accommodate diverse misfolded or folded substrates with different dimensions. The structure of DNAJC5 oligomers may shed light on the principle of this folding-independent translocation pathway.

## Materials and methods

**Key resources table**

| Reagent type (species) or resource | Designation | Source or reference | Identifiers | Additional information |
|---|---|---|---|---|
| Antibody | Mouse monoclonal anti-α-synuclein | BD Biosciences | Cat# 610787 | (1:500) |
| Antibody | Rabbit polyclonal anti-α-synuclein | Proteintech | Cat# 10842-1-AP | (1:500) |
| Antibody | Rabbit polyclonal anti-DNAJC5 | RayBiotech | Cat# 144-10489-200 | (1:1,000) |
| Antibody | Mouse monoclonal anti-alpha tubulin | Abcam | Cat# ab7291 | (1:2,000) |
| Antibody | Mouse monoclonal anti-Alix | Santa Cruz Biotechnology | Cat# Sc-53540 | (1:1,000) |
| Antibody | Rabbit monoclonal anti-CD9 | Cell Signaling Technology | Cat# 13174S | (1:1,000) |
| Antibody | Mouse monoclonal anti-PDI | Enzo Life Sciences | Cat# ADI-SPA-891-D | (1:1,000) |
| Antibody | Mouse monoclonal anti-CD63 | Thermo Fisher Scientific | Cat# BDB556019 | (1:1,000) |
| Antibody | Mouse monoclonal anti-Flotillin-2 | BD Biosciences | Cat# 610383 | (1:1,000) |
| Antibody | Mouse monoclonal anti-Transferrin Receptor | Thermo Fisher Scientific | Cat# 13-6800 | (1:1,000) |
| Antibody | Mouse monoclonal anti-GM130 | BD Biosciences | Cat# 610823 | (1:1,000) |
| Antibody | Rabbit monoclonal anti-Tom20 | Cell Signaling Technology | Cat# 42406S | (1:1,000) |
| Antibody | Rabbit polyclonal anti-GFP | Fisher Scientific | Cat# NC9589665 | (1:1,000) |
| Antibody | Rabbit polyclonal anti-LC3B | Novus Biologicals | Cat# NB100-2220 | (1:1,000) |
| Antibody | Rabbit monoclonal anti-Citrate Synthase | Cell Signaling Technology | Cat# 14309S | (1:1,000) |
| Antibody | Rabbit polyclonal anti-Dopamine transporter | Bioss Antibodies | Cat# BS-1714R | (1:1,000) |
| Antibody | Rabbit monoclonal anti-beta III Tubulin | Abcam | Cat# ab215037 | (1:1,000) |

*Continued on next page*

*Continued*

| Reagent type (species) or resource | Designation | Source or reference | Identifiers | Additional information |
|---|---|---|---|---|
| Antibody | Rabbit polyclonal anti-Tyrosine hydroxylase | Millipore | Cat# AB152 | (1:1,000) |
| Antibody | Chicken polyclonal Microtubule-associated protein 2 | Abcam | Cat# ab92434 | (1:1,000) |
| Antibody | Mouse monoclonal anti-FLAG | Sigma-Aldrich | Cat# F9291 | (1:1,000) |
| Strain, strain background (*Escherichia coli*) | XL1-Blue competent cells | MacroLab Berkeley | N/A | |
| Strain, strain background (*E. coli*) | Rossetta (DE3) pLysS competent cells | MacroLab Berkeley | N/A | |
| Chemical compound, drug | Anti-FLAG M2 Affinity Gel | Sigma-Aldrich | Cat# A2220-5ML | |
| Chemical compound, drug | Dimethyl sulfoxide (DMSO) | Thermo Fisher Scientific | Cat# BP231-100 | |
| Chemical compound, drug | Quercetin | Sigma-Aldrich | Cat# Q4951-10G | |
| Chemical compound, drug | 2-Bromopalmitic acid | Millipore Sigma | Cat# 21604-1G | |
| Chemical compound, drug | Balfilomycin A1 | Cayman Chemical | Cat# 11038 | |
| Chemical compound, drug | Retinoic acid | Sigma-Aldrich | Cat# R2625-100MG | |
| Chemical compound, drug | HaloTag Oregon Green Ligand | Promega | Cat# G2802 | |
| Chemical compound, drug | HaloTag TMR Ligand | Promega | Cat# G8251 | |
| Chemical compound, drug | Prolong Gold with DAPI | Thermo Fisher Scientific | Cat# P36931 | |
| Chemical compound, drug | Proteinase K | Sigma-Aldrich | Cat# P2308 | |
| Peptide, recombinant protein | α-syn tandem repeats protein | This paper | N/A | |
| Commercial assay or kit | Nano-Glo Luciferase Assay System | Promega | Cat# N1150 | |
| Commercial assay or kit | LEGEND MAX Human α-synuclein (Colorimetric) ELISA Kit | BioLegend | Cat# 448607 | |
| Commercial assay or kit | Mouse α-synulcein ELISA Kit | Abcam | Cat# ab282865 | |
| Cell line (*Homo sapiens*) | HEK293T cells | Cell Culture Facility, UC Berkeley | N/A | |
| Cell line (*H. sapiens*) | HEK293-lenti-X cells | Cell Culture Facility, UC Berkeley | N/A | |
| Cell line (*H. sapiens*) | MDA-MB-231 cells | Cell Culture Facility, UC Berkeley | N/A | |
| Cell line (*H. sapiens*) | Hela cells | Cell Culture Facility, UC Berkeley | N/A | |
| Cell line (*H. sapiens*) | SH-SY5Y cells | Cell Culture Facility, UC Berkeley | N/A | |
| Cell line (*H. sapiens*) | U2OS cells | Cell Culture Facility, UC Berkeley | N/A | |
| Cell line (*H. sapiens*) | HEK293T-DNAJC5-CRISPR KO cells | This study | N/A | |
| Cell line (*H. sapiens*) | SH-SY5Y-DNAJC5-shRNA KD cells | This study | N/A | |
| Cell line (*Mus musculus*) | Mouse Embryonic Stem Cells | | | |
| Cell line (*M. musculus*) | Mouse Embryonic Stem Cells-hDNAJC5-OE cells | This study | N/A | |
| Cell line (*H. sapiens*) | Human-Induced Pluripotent Stem Cells | University of Oxford; EBiSC repository | N/A | |
| Recombinant DNA reagent | mCherry-Rab5CA (Q79L) | Addgene | Cat# 35138 | |
| Recombinant DNA reagent | BFP-Rab5CA (Q79L) | This study | N/A | |
| Recombinant DNA reagent | SNCA (Myc-DDK-tagged)-Human synuclein, alpha | OriGene Technoogy | Cat# RC221446 | |

*Continued on next page*

*Continued*

| Reagent type (species) or resource | Designation | Source or reference | Identifiers | Additional information |
|---|---|---|---|---|
| Recombinant DNA reagent | CSP (DNAJC5) (NM_025219) Human Tagged ORF Clone | OriGene Technology | Cat# RC208826 | |
| Recombinant DNA reagent | TCH1003-MGC premier cDNA clone for USP19 | transOMIC | Cat# TCH1003 | |
| Recombinant DNA reagent | pCMV-α-synuclein-A30P | Gift of Dr. Thomas Südhof lab | N/A | |
| Recombinant DNA reagent | pCMV-α-synuclein-E46K | Gift of Dr. Thomas Südhof lab | N/A | |
| Recombinant DNA reagent | pCMV-α-synuclein-A53T | Gift of Dr. Thomas Südhof lab | N/A | |
| Recombinant DNA reagent | αS-2 (α-syn-α-syn) tandem dimer | Gift of Michael Woodside lab | N/A | |
| Recombinant DNA reagent | αS-4 (α-syn-α-syn-α-syn-α-syn) tandem Tetramer | Gift of Michael Woodside lab | N/A | |
| Recombinant DNA reagent | pCMV-αS-2 | This study | N/A | |
| Recombinant DNA reagent | pCMV-αS-4 | This study | N/A | |
| Recombinant DNA reagent | pCMV-DNAJC5-L115R | This study | N/A | |
| Recombinant DNA reagent | pCMV-DNAJC5-L116Δ | This study | N/A | |
| Recombinant DNA reagent | pCMV-DNAJC5 (WT)-HaloTag | This study | N/A | |
| Recombinant DNA reagent | pCMV-DNAJC5 (L115R)-HaloTag | This study | N/A | |
| Recombinant DNA reagent | pCMV-XPACK-DNAJC5 (L115R)-HaloTag | This study | N/A | |
| Recombinant DNA reagent | mNeonGreen-α-synuclein | This study | N/A | |
| Recombinant DNA reagent | 3H-α-synuclein | This study | N/A | |
| Recombinant DNA reagent | α-synuclein-3H | This study | N/A | |
| Recombinant DNA reagent | pCMV-XPACK-DNAJC5 (WT) | This study | N/A | |
| Recombinant DNA reagent | pCMV-XPACK-DNAJC5 (L115R) | This study | N/A | |
| Recombinant DNA reagent | pCMV-DEAD XPACK-DNAJC5 (L115R) | This study | N/A | |
| Recombinant DNA reagent | pCMV-DNAJC5-ΔJ (Δ14–82) | This study | N/A | |
| Recombinant DNA reagent | pCMV-DNAJC5-ΔC10 (D189–198) | This study | N/A | |
| Recombinant DNA reagent | pCMV-DNAJC5-ΔC20 (D179–198) | This study | N/A | |
| Recombinant DNA reagent | pCMV-DNAJC5-ΔC30 (D169–198) | This study | N/A | |
| Recombinant DNA reagent | pCMV-DNAJC5-ΔC40 (D151–198) | This study | N/A | |
| Recombinant DNA reagent | pCMV-pOTC-GFP | This study | N/A | |
| Recombinant DNA reagent | pCMV-pOTC-3H-GFP | This study | N/A | |
| Recombinant DNA reagent | pX330-Venus-DNAJC5-Exon 4-gRNA | This study | N/A | |
| Recombinant DNA reagent | pLenti-CMV-DNAJC5 | This study | N/A | |
| Recombinant DNA reagent | pLenti-CMV-DNAJC5 (L115R) | This study | N/A | |
| Recombinant DNA reagent | pIKO.1-DNAJC5-ShRNA | This study | N/A | |
| Software, algorithm | Fiji (ImageJ) | NIH | https://imagej.nih.gov/ij/ | |
| Software, algorithm | PyMOL | Schrödinger | https://pymol.org/2/ | |
| Software, algorithm | Prism 8 | Graphpad | https://www.graphpad.com/scientific-software/prism/ | |
| Software, algorithm | AlphaFold Protein Structure Database | DeepMind | https://alphafold.ebi.ac.uk/ | |

## Cell culture and transfection

All immortalized cell lines were obtained from the UC-Berkeley Cell Culture Facility and were confirmed by short tandem repeat (STR) profiling and tested negative for mycoplasma contamination. Cells were grown at 37°C in 5% $CO_2$ and maintained in Dulbecco's modified Eagle's medium (DMEM) supplemented with 10% fetal bovine serum (FBS). For secretion assays, the FBS concentration was reduced to 1% for up to 36 hr during which time the growth rate of cells slowed but cells remained viable. For EV preparation and medium fractionation, we grew cells in DMEM supplemented with exosome-depleted FBS. Exosome-depleted FBS was prepared by overnight centrifugation of 30% diluted FBS in DMEM at 100,000×$g$. Transfection of plasmids into cells was performed using Lipofectamine 2000 (Thermo Fisher Scientific, Waltham, MA) according to the manufacturer's protocols.

## Reconstitution of α-syn secretion in HEK293T cells

HEK293T cells (cultured in six-well plates) were cultured to 60% confluence and co-transfected with plasmids encoding different constructs of α-syn and DNAJC5. pCMV-GFP was used as a transfection control in all secretion experiments. At 4 hr after transfection, we replaced cell culture medium with DMEM supplemented with 1% FBS containing indicated drugs for treatment. At indicated time points, we collected media fractions which were centrifuged at 1000×$g$ for 10 min to remove floating cells and cell debris. The media were mixed with methanol/chloroform to precipitate proteins which were collected by centrifugation (10,000×$g$ × 10 min) and resuspended in SDS-PAGE sample loading buffer to achieve concentration (20-fold). Cells were lysed in lysis buffer (10 mM Tris, pH 7.4, 100 mM NaCl, and 1% Triton X-100). Both the concentrated media and cell lysate fractions were analyzed by immunoblot.

To exclude the release of cytoplasmic proteins from cell death, we monitored the viability of cells after transfection with a Countess II Automated Cell Counter (Thermo Fisher Scientific) using trypan blue staining.

For a nanoluciferase-based assay, media fractions were collected and centrifuged at 1000×$g$ for 10 min. The supernatant fractions were harvested and further diluted with PBS buffer (1000-fold). The nanoluciferase activity was assayed using a Nano-Glo Luciferase Assay System (Promega, Madison, WI) according to the manufacturer's protocol.

## Membrane and cytosol fractionation

Cells (one 10 cm dish) were cultured to 70% confluence and transfected with different constructs of DNAJC5. One day after transfection, we harvested the transfected cells by scraping in 1 ml B88 (20 mM HEPES-KOH, pH 7.2, 250 mM sorbitol, 150 mM potassium acetate, and 5 mM magnesium acetate) plus a cocktail of protease inhibitors (Sigma-Aldrich, St. Louis, MO). Cells were homogenized by 10 passages through a 22G needle. Homogenates were centrifuged at 500×$g$ for 10 min and the resulting post-nuclear supernatant (PNS) fractions were centrifuged at 100,000×$g$ for 1.5 hr. High-speed supernatant fractions were then subjected to a repeat centrifugation to achieve a clarified cytosol fraction. The pellet fraction was washed and resuspended in the same volume of B88. Resuspended material was also centrifuged again to collect a washed membrane fraction. Membranes were lysed in lysis buffer.

For membrane fractionation, the PNS was subjected to differential centrifugation at 3000×$g$ (10 min), 25,000×$g$ (20 min), and 100,000×$g$ (30 min). Membrane fractions were normalized to phosphatidylcholine content and analyzed by immunoblot (*Ge et al., 2013*).

For proteinase K protection assays, the 25,000×$g$ membrane fraction was aliquoted into three tubes: one without proteinase K, one with proteinase K (10 μg/ml), and one with proteinase K plus TritonX-100 (0.5%). The incubation was conducted on ice for 20 min and stopped by sequential addition of PMSF (1 mM) and sample buffer and samples were then heated on metal block at 95°C for 5 min and analyzed by SDS-PAGE and immunoblot.

## In vitro depalmitoylation assay

Cells (HEK293T, MDA-MB-231, or Hela) were transfected with DNAJC5. Cellular membranes were prepared as described above. For chemical depalmitoylation, the membranes were resuspended and incubated with 0.5 M hydroxylamine (pH 7.2) or 0.5 M Tris (pH 7.2, control) at room temperature

overnight in the presence of a cocktail of protease inhibitors (Sigma-Aldrich, St. Louis, MO). The mobility of DNAJC5 was examined by SDS-PAGE followed by immunoblot.

## CRISPR/Cas9 genome editing

gRNA targeting exon 4 of DNAJC5 (CACCGGAGGCCGCAGAAGACAAACA) was inserted into a pX330-based plasmid expressing Venus fluorescent protein (*Shurtleff et al., 2016*). HEK293T cells were transfected with pX330-pX330-Venus-DNAJC5-Exon 4-gRNA by Lipofectamine 2000 (Thermo Fisher Scientific, Waltham, MA). After 48 hr, we diluted the cells and single colonies were isolated, expanded, and determined for DNAJC5 KO by immunoblot.

## Medium fractionation and EV preparation

Conditioned medium was harvested and centrifuged first at 1500×$g$ for 20 min followed by 10,000×$g$ for 30 min and 100,000×$g$ for 1.5 hr. The supernatant fractions at each step were collected and treated with methanol/chloroform to precipitate proteins which were then collected by centrifugation. Pellet fractions were resuspended in sample buffer to achieve a 20-fold concentration. The sedimented fractions at each step were also collected and resuspended in sample buffer. All the fractions were analyzed by immunoblot.

EVs were isolated by buoyant density flotation on a sucrose step gradient. The pellet fraction from a 100,000×$g$ centrifugation was resuspended in PBS and mixed with 60% sucrose buffer (10 mM Tris-HCl pH 7.4, 100 mM NaCl) to achieve a final sucrose concentration >50% as measured with a refractometer. Aliquots of 40% (5 ml) and 10% (2 ml) sucrose buffer were sequentially overlaid above the sample. The tubes were then centrifuged at 150,000×$g$ for 16 hr in an SW41 Ti swinging-bucket rotor (Beckman Coulter). After centrifugation, 0.5 ml fractions were collected from top to bottom and samples were analyzed by SDS-PAGE and immunoblot.

## Co-immunoprecipitation

Media fractions were collected and centrifuged at 1000×$g$ for 10 min. The supernatant fractions were collected and concentrated (20-fold) using a 10 kDa Amicon filter (Millipore, Billerica, MA). Concentrated media fractions (1 ml) were incubated with 20 µl of anti-FLAG M2 affinity gel (Sigma-Aldrich, St. Louis, MO) for 1 hr at 4°C. After washing 5× with lysis buffer, SDS-PAGE sample loading buffer was added to the beads and samples were processed for SDS-PAGE and immunoblot.

## Protein purification

The purification of different α-syn tandem-oligomer constructs was performed as previously described (*Dong et al., 2018*). Briefly, an osmotic shock protocol was adapted to enrich proteins released from the periplasm of transfected *Escherichia coli*. The supernatant fraction containing released proteins was subjected to ammonium sulfate (AS) precipitation, with 50%, 45%, and 40% saturated concentration of AS for monomer, dimer, and tetramer, respectively. After overnight precipitation, the precipitated proteins were collected by centrifugation at 100,000×$g$ for 30 min. The pellet fractions were dissolved in Buffer A (20 mM Tris-HCl pH 8.0) and clarified by repeated centrifugation at 100,000×$g$ for 30 min.

Clarified supernatants were applied to an equilibrated HiPrep Q Fast Flow 16/10 column (GE Healthcare, Chicago, IL). Eluted proteins were collected, concentrated by 10 kDa Amicon filter (Millipore, Billerica, MA) and further purified by gel filtration (Superdex-200, GE Healthcare) with PBS used as gel filtration buffer. Purified proteins were assessed by SDS-PAGE followed by coomassie-blue staining.

## IF and live-cell Imaging

For IF, U2OS cells were washed once with PBS and immediately fixed by 4% EM-grade paraformaldehyde (Electron Microscopy Science, Hatfield, PA) for 10 min at room temperature. Cells were washed three times with PBS and blocked and permeabilized for 30 min in permeabilization buffer (5% FBS and 0.1% saponin in PBS). hiPSC dopamine neurons were fixed with 4% paraformaldehyde in PBS and 0.1% Triton-X was used for permablization (10 min) followed by blocking in 10% normal donkey serum for 1 hr. Cells were then incubated with 1:100 dilution of primary antibodies overnight at 4°C. After three washes with PBS, cells were incubated with 1:500 dilution of fluorophore-conjugated secondary

antibodies for 30 min at room temperature. Prolong Gold with DAPI (Thermo Fisher Scientific) was used as mounting solution. Images were acquired with a Zeiss LSM900 confocal microscope and analyzed with Fiji/ImageJ software (https://imagej.nih.gov/ij/).

For live-cell imaging, cells were cultured in 35 mm glass bottom dishes (MatTek). The addition of HaloTag fluorescent ligands were added according to the manufacturer's protocol (Promega). After incubation, the medium was replaced with Opti-MEM supplemented with 10% FBS. Imaging was performed using a Zeiss LSM900 confocal microscope in a temperature-controlled (37°C and 5% $CO_2$) environment.

## Mitochondria purification

HEK293T cells were trypsinized and collected by centrifugation. Cells were washed twice with NKM buffer (1 mM Tris HCl, pH7.3, 0.13 M NaCl, 5 mM KCl, and 7.5 mM $MgCl_2$), and resuspended in six packed cell volumes of homogenization buffer (10 mM Tris pH 7.4, 10 mM KCl, and 0.15 mM $MgCl_2$). Cells were homogenized by 10 passages through a 22G needle. Cell homogenates were mixed gently with the same volume of 2.3 M sucrose solution and centrifuged at 1200×$g$ for 5 min to remove unbroken cells and large cell debris. The recovered supernatant fractions were centrifuged at 7000×$g$ for 10 min. Mitochondria enriched in the pellet fraction were resuspended in three packed cell volumes of Mitochondria Suspension Buffer (10 mM Tris, pH 7.3, 0.15 mM MgCl2, and 0.25 mM sucrose).

## Differentiation of SH-SY5Y cells

SH-SY5Y neuroblastoma cells were maintained in DMEM supplemented with 1× nonessential amino acid (NEAA), 1× sodium pyruvate, and 10% FBS. Differentiation was induced by lowering the FBS in culture medium to 1% plus 10 µM RA. Cell medium was replaced each 3 days to replenish RA. Cell morphology was monitored by microscopy and experiments on SH-SY5Y cells were performed from D6 of differentiation.

## shRNA knockdown

plKO.1-Hygro plasmids-containing shRNA targeting DNAJC5 (ccggGCAACCTCAGATGACATTAAACTCGAGTTTAATGTCATCTGAGGTTGCTTTTTG) together with pMD2.G and PsPAX2 were transfected into HEK293T cells to produce lentiviral particles for 72 hr. Lentivirus particles were concentrated with Lenti-X Concentrator (Takara Bio). SH-SY5Y was transduced by lentivirus before differentiation. Three days post transduction, cells were selected with 250 µg/ml hygromycin for 10 days. The selected cells were differentiated, and the knockdown was verified with immunoblot.

## Culture and differentiation of mESCs

Mouse ESCs (R1) were maintained and differentiated into dopaminergic neurons following a modified protocol from *Ni et al., 2013*. Briefly, R1 cells were maintained in a feeder-independent system, plated in 0.1% gelatin (StemCell Technologies) and cultured in KSR medium consisting of KnockOut DMEM, 20% KnockOut serum replacement, 2 mM L-glutamine, 0.1 mM NEAAs, 0.1 mM β-mercaptoethanol, and 1000 U/ml leukemia inhibitory factor (LIF, Chemicon International) with a media change every day. Cells were then grown in aggregate cultures to form EBs in DMEM/F12 media supplemented with 10% knockout serum replacement, 2.4% N2, 4500 mg/L Glucose, 2 mM L-glutamine, and 0.1 mM β-mercaptoethanol. EBs were formed for 4 days and then plated on 10 µg/ml laminin-coated plates. After 24 hr of culture, the media were replaced by DMEM/F12, 3% KO serum, N2, Glucose, 1× Glutamine, and 2-BME supplemented with 1% Insulin/Transferrin/Selenium with a media change every day. After 7 days, cells were dissociated by Accutase StemPro and plated on laminin-coated plates using a 1:1 ratio of Neurobasal media and DMEM/F12, N2, B27, 2 mM L-glutamine, 0.1 mM NEAAs, 0.1 mM β-mercaptoethanol supplemented with 20 ng/ml bFGF (R&D Systems), 200 ng/ml SHH (R&D Systems), and 25 ng/ml FGF8b (R&D Systems) with a media change every day. After 8 days, the culture medium was changed to Neurobasal/B27 medium supplemented with 0.5 mM dbcAMP (Santa Cruz Biotechnology), 0.2 mM ascorbic acid (StemCell Technologies), 20 ng/ml BDNF (Petrotech), and 20 ng/ml GDNF (Petrotech) with a media change every other day.

## Differentiation and culture of dopaminergic neurons from hiPSCs

Primary fibroblasts derived from PD patients carrying the *GBA-N370S* mutation and a healthy control (below) were reprogrammed to pluripotency as described previously (*Fernandes et al., 2016*) and

clones were selected, tested for mycoplasma and QCed according to established protocols (*Lang et al., 2019*). hiPSCs were differentiated toward dopaminergic fate as described by *Kriks et al., 2011* with small modifications (*Beevers et al., 2017*). Briefly, hiPSCs were patterned for 21 days with a growth factor cocktail to promote differentiation toward ventral midbrain neuronal progenitor cells for 11 days (10 mM SB431542, Tocris; 100 nM LDN193189, Sigma-Aldrich; 2 mM puromorphamine, Millipore; 100 ng/ml sonic hedgehog, Bio-Techne; 100 ng/ml fibroblast growth factor-8a, Bio-Techne, and 3 mM CHIR99021), followed by 10 days of differentiation to dopaminergic neurons (20 ng/ml brain-derived neurotrophic factor, Peprotech; 20 ng/ml glial cell line-derived neurotrophic factor, Peprotech; 1 ng/ml transforming growth factor type β3, Peprotech; 0.5 mM dibutyryl cAMP, Sigma-Aldrich; 0.2 mM Ascorbic acid, Sigma-Aldrich, and 10 mM DAPT, Abcam). Neurons were matured for a further 2 weeks (to day 35) for α-synuclein secretion, or for a further 5 weeks (to day 65) for analysis by SDS-PAGE and immunoblot. Neurons were then treated with DMSO, quercetin, or 2-bromoplamitic acid for 3 days.

| Donor ID | Study ID | Genotype | Age/gender | Characterization |
|---|---|---|---|---|
| SFC156-03 | Control | GBA wt/GBA wt | 75 Male | *Lang et al., 2019* |
| MK071 | PD 1 | GBA N370S/GBA wt | 81 Female | *Fernandes et al., 2016* |
| MK088 | PD 2 | GBA N370S/GBA wt | 46 Male | *Fernandes et al., 2016* |
| SFC871-03-09 | PD 3 | GBA N370S/GBA wt | 70 Female | *Bogetofte et al., 2021* |
| MK082 | PD 4 | GBA N370S/GBA wt | 51 Male | *Lang et al., 2019* |

## Extracellular α-syn measurements

Commercial ELISA Kits as listed in Key Resources Table were used to quantify the extracellular α-syn in SH-SY5Y and mouse mDA neuronal culture. Conditioned media were collected and centrifuged at 1000×*g* for 10 min. α-syn in the recovered supernatant was measured using the protocol provided in the kit.

α-syn secretion by hiPSC-derived dopaminergic neurons was measured as described previously (*Fernandes et al., 2016*; *Fernandes et al., 2016*) using an electro-chemiluminescent assay (Meso Scale Discovery, MD, Cat# K151TGD-2) and a MESO QuickPlex SQ 120 instrument (Meso Scale Discovery) according to the manufacturer's instructions. Briefly, culture media were collected 3 days after drug treatment (differentiation day 38) and quantified relative to a standard curve. Data were normalized relative to the total protein content of the cells from which the conditioned media had been collected, as determined by BCA assay.

## Immunoblots

Cell lysate, cytosol, or membrane samples, and EV samples were mixed with SDS sample loading buffer. Samples were heated at 95°C for 5 min and separated on SDS-PAGE gels. Proteins were transferred to PVDF membranes (EMD Millipore, Darmstadt, Germany), blocked with 5% bovine serum albumin in TBST (20 mM Tris pH 7.4, 150 mM NaCl, and 0.1% Tween-20) and incubated overnight with primary antibodies. For immunoblots from hiPSC dopamine neurons, samples in loading buffer were heated to 70°C for 10 min and blocking was carried out with 5% skimmed milk. For immunoblots of endogenous α-syn in SH-SY5Y cells, PVDF membranes were fixed with 0.4% paraformaldehyde (Electron Microscopy Science, Hatfield, PA) in TBST at room temperature for 30 min (*Lee and Kamitani, 2011*). Blots were then washed with TBST, followed by incubation with anti-rabbit or anti-mouse secondary antibodies (GE Healthcare Life Sciences, Pittsburgh, PA). Detection was performed with Supersignal Chemiluminescent substrate (Thermo Fisher Scientific) and quantified with Fiji/ImageJ. Primary antibodies used in this study were listed in Key Resources Table. All antibodies used for immunoblots were diluted 1:1000, except for 1:2000 of mouse anti-Tubulin, 1:500 of rabbit anti-α-syn and of rabbit anti-tyrosine hydroxylase.

## Acknowledgements

The authors thank Dr. Michael Woodside and Dr. Thomas Südhof for sharing the plasmids, and Dr. Kalina Naidoo, Dr. Mootaz Salman and William McGuinness for preparation of iPSC samples. The

authors also thank the staff at the UC Berkeley Shared Facilities, the Cell Culture Facility, the DNA Sequencing Facility, and the Biological Imaging Facility. SW is supported as Associate of the HHMI. RS is an Investigator of the HHMI, a senior fellow of the UC Berkeley Miller Institute of Science and the Scientific Director of Aligning Science across Parkinson's disease.

## Additional information

### Competing interests

Randy Schekman: Reviewing editor, *eLife*. The other authors declare that no competing interests exist.

### Funding

| Funder | Grant reference number | Author |
| --- | --- | --- |
| Howard Hughes Medical Institute | | Randy Schekman |
| NIH Biology and Biotechnology of Cell and Gene Therapy Training Program | NIH training program T32GM139780 | Nancy C Hernandez Villegas |
| Aligning Science Across Parkinson's | ASAP-020370 | Richard Wade-Martins Randy Schekman |

The funders had no role in study design, data collection and interpretation, or the decision to submit the work for publication.

### Author contributions

Shenjie Wu, Daniel W Sirkis, Conceptualization, Formal analysis, Investigation, Methodology, Writing - original draft, Writing - review and editing; Nancy C Hernandez Villegas, Formal analysis, Funding acquisition, Investigation, Methodology, Writing - original draft, Writing - review and editing; Iona Thomas-Wright, Formal analysis, Investigation, Writing - original draft, Writing - review and editing; Richard Wade-Martins, Formal analysis, Funding acquisition, Writing - original draft, Writing - review and editing; Randy Schekman, Conceptualization, Formal analysis, Supervision, Funding acquisition, Investigation, Writing - original draft, Writing - review and editing

### Author ORCIDs

Shenjie Wu http://orcid.org/0000-0001-7547-0048
Nancy C Hernandez Villegas http://orcid.org/0000-0002-3982-7553
Daniel W Sirkis http://orcid.org/0000-0003-3440-8859
Randy Schekman http://orcid.org/0000-0001-8615-6409

### Decision letter and Author response

Decision letter https://doi.org/10.7554/eLife.85837.sa1
Author response https://doi.org/10.7554/eLife.85837.sa2

## Additional files

### Supplementary files

• MDAR checklist

### Data availability

All data generated or analysed during this study are included in the manuscript and supporting file; Source Data files have been provided for all the Figures.

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
