## [Editor Report]

This study describes a comprehensive set of classical techniques in biochemistry and cell biology characterizing the unconventional secretory mechanism of α-synuclein, an important contributor to neurodegeneration. The major finding is that palmitoylated DNAJC5 oligomers play a central role in this unusual secretory pathway, presumably by mediating entry of α-synuclein into late endosomes. Future work will clarify the mechanisms by which this cargo is exported from the cell.

---

## [Decision Letter]

**Decision letter after peer review:**

[Editors’ note: the authors submitted for reconsideration following the decision after peer review. What follows is the decision letter after the first round of review.]

Thank you for submitting the paper "Unconventional secretion of α-synuclein mediated by palmitoylated DNAJC5 oligomers" for consideration by *eLife*. Your article has been reviewed by 3 peer reviewers, and the evaluation has been overseen by a Senior Editor. The reviewers have opted to remain anonymous.

Comments to the Authors:

We are sorry to say that the reviewers agreed that too much additional information would be needed to continue consideration of the present manuscript version. However, we would welcome a new submission that can address the most substantive issues raised by the reviewers as described below. The reviewers also asked me to highlight a concern related to the potential impact of the story in terms of the limitations of over-expression in a single cell line that may not be relevant to synucleinopathies (versus neurons, for instance).

*Reviewer #1 (Recommendations for the authors):*

In this paper, the precise role of DNAJC5 in unconventional secretion of α-synuclein was studied. Elucidating the molecular mechanism of this process is not only an interesting question in basic research but is also of high relevance with regard to potential biomedical applications in the context of neurodegenerative disorders. The current state of the literature is quite heterogenous with a number of distinct pathways being proposed to play a role in α-synuclein secretion, including a potential role for exosomes and cell-connecting nanotubes, among others.

Here, the authors aimed at reconstituting α-synuclein secretion in a general human cell line, HEK293T cells. A particular focus was given to DNAJC5, a protein that has previously been demonstrated to play a role in this process. In addition, palmitoylation was already known to mediate membrane attachment of DNAJC5. The authors demonstrate that both palmitoylated and non-palmitoylated DNAJC5 are secreted alongside α-synuclein. However, while palmitoylated DNAJC5 was found associated with some kind of extracellular vesicles, non-palmitoylated DNAJC5 and monomeric α-synuclein were found as soluble proteins in cellular supernatants. The authors further show that palmitoylation of DNAJC5 is required for α-synuclein secretion.

Using imaging techniques, the authors found DNAJC5 and α-synuclein to be associated with a population of endosomes. Along with cell fractionation and protease-protection experiments, these experiments were interpreted in a way that both proteins are located in the lumen of endosomes. They were further taken as evidence that DNAJC5 mediates translocation of α-synuclein from the cytoplasm into the lumen of endosomes.

In a further set of experiments, the authors aimed at addressing the folding state of α-synuclein while it is transported into the extracellular space. Using an elegant experimental system, evidence was provided that α-synuclein remains folded during all steps of its transport route.

Finally, the authors provide evidence for DNAJC5 to oligomerize, a process that is enhanced when the J domain is deleted. They also find that the J domain is not required for α-synuclein secretion.

As a bottom line the authors suggest that palmitoylated species of DNAJC5 oligomers form or link to other components of a pore in late endosomes through which α-synuclein translocates into their lumen. This potential step of α-synuclein membrane translocation is proposed to be the critical process that targets α-synuclein into the extracellular space.

In general, I appreciate this study that uses classical methods in biochemistry and cell biology collecting a number of important insights about the unconventional secretory pathway of α-synuclein.

In the following, I would like to point out a number of questions and suggestions the authors may want to address to improve this manuscript:

– Why and how do cells secrete non-palmitoylated DNAJC5, a variant that cannot associate with membranes? And why does palmitoylated DNAJC5 end up on extracellular vesicles, a species that is proposed to be inserted as an oligomer into the limiting membrane of late endosomes? When late endosomes fuse with the plasma membrane as suggested by the authors, palmitoylated DNAJC5 should remain in the inner leaflet of the plasma membrane. Based on the mechanism of α-synuclein secretion proposed in this study, both of these phenomena are difficult to incorporate into the concept of the authors.

– I am not sure whether the imaging experiments shown in Figure 4 and its supplements unequivocally demonstrate a luminal localization of the proteins in question. I also find the protease protection experiments to be limited in their conclusiveness since only a minor fraction of α-synuclein and DNAJC5 appear to be protected. Is there a way to produce further independent evidence for these claims? Finally, the time-lapse video provided as a supplement does not convince me with regard to the suggested role of DNAJC5 promoting membrane translocation of α-synuclein.

– To me it seems unclear whether the populations of α-synuclein and DNAJC5 found on endosomes are necessarily linked to the unconventional secretory mechanism of α-synuclein. Isn't it also possible that this material is derived from endocytosis of α-synuclein and DNAJC5? In particular with regard to the populations that are potentially in the lumen of these endosomes?

– The experimental evidence for DNAJC5 oligomers mediating membrane translocation of α-synuclein remains limited. I think this paper could contribute a major advance to the field if the observed oligomers could be characterized in more detail as has been done in other studies in the field. For example, beyond the observation of SDS-resistant oligomers in gels, FCS techniques could be used to precisely determine the oligomeric state of DNAJC5 in a native state. In addition. there are biophysical assays that allow to observe pore formation with purified proteins and artificial membrane vesicles. If there is a pore, its cut-off size could be determined and compared to the dimensions of α-synuclein. The authors indicate in the discussion that DNAJC5 might not be able to form a pore itself and that this process might be coupled to CD98. I think these aspects remain vague, however, insights into these questions would make up a true advance in the field.

– While the technique used to study the folding state of α-synuclein during its transport into the extracellular space is very elegant, in my opinion a positive control is missing, i.e. an unconventionally secreted protein whose secretion depends on an intermediate state that is unfolded. I think Interleukin 1β could serve as such a control.

*Reviewer #2 (Recommendations for the authors):*

In this manuscript, Wu et al. investigate the mechanisms through which the co-chaperone DNAJC5 regulates unconventional secretion of α-synuclein. The authors have some interesting findings such as palmitoylation is important for this process, but whether DNAJC5 itself was necessary is unclear. Addressing this ambiguity could lead to meaningful insights and the study could provide a cell biological basis for α-synuclein spread in Parkinson's disease.

Several additional experiments and quantification of the data presented will greatly enhance the quality of this manuscript.

Figure 1-NP-DNAJC5 and P-DNAJC5. The palmitoylation status of P-DNAJC5 should be confirmed by treating with hydroxylamine and checking if the protein now runs at NP-DNAJC5.

Figure 1-Please rule out that expression of the various constructs is not causing cell death.

Figure 1-S1-Can the authors comment on why DNAJC5 levels in the medium is decreased when PD mutants of α-synuclein are expressed? It would be helpful to quantitate these data to see the impact of PD mutations.

Figure 1-S2-Quercetin is rather non-specific and toxic. The data would be bolstered if in Figure S2E, the luciferase data is accompanied by western blots for α-synuclein and DNAJC5 (in addition to tubulin).

Figure 1-S2F-G- The effect of the KD should be quantified. KD of DNJAC5 does not seem to impact α-synuclein secretion. Is the effect on α-synuclein secretion even significant? These data with Figure 3 would argue that DNAJC5 itself is not necessary for secretion of α-synuclein. In this regard, can you substitute DNAJC5 with any heavily palmitoylated protein?

Figure 1-S2F-G- These data suggest that PATs that palmitoylate DNAJC may be important for secretion of α-synuclein. Also, KD of HIP14 would be instructive.

Figure 3-the ratio of DNAJC5-NP/P for the various conditions should be quantified.

Figure 3-The L115R mutant is retained in the Golgi. Does treatment with 2-BA also result in retention of DNAJC5 in the Golgi? If so, how does this fit into the model proposed for α-synuclein secretion.

Figure 4-How many large endosomes contain both DNAJC5 and α-synuclein? This is important to ascertain as it is not evident from the images shown that the miniscule amount of α-synuclein in endosomes accounts for amounts seen by western blotting of media. Also, why is the α-synuclein within the endosome not diffuse (as would be expected from a monomer) but rather punctate?

Figure 4-mNeo-Green tagging in the N-terminus of α-synuclein will prevent the protein from folding properly.

Lastly recapitulating these data in neurons would increase the significance of these findings.

Discussion is very speculative.

*Reviewer #3 (Recommendations for the authors):*

The authors describe a mechanism by which DNAJC5 stimulates the extracellular secretion of α-synuclein in a manner dependent on the palmitoylation of DNAJC5 which serves to anchor the protein to the membrane. There may be an additional role for oligomerization of DNAJC5 in facilitating α-syn secretion, as modifications to the protein which induce oligomerization appear to rescue deficits observed in palmitoyl-deficient mutants. The authors propose that the secretion of α-synuclein proceeds through an intermediate step of localization into late endosomal compartments, which appears to be disrupted in DNAJC5 mutants. Broadly this paper uses a combination of cellular fractionation, immunoblot and immunofluorescence microscopy in HEK 293 cells overexpressing a range of DNAJC5 constructs to dissect the role of this protein in α-synuclein export.

Advance over previous work:

The observation that DNAJC5 can mediate extracellular secretion of aSyn was previously described (Fontaine et al., 2016). Similarly, the importance of palmitoylation of DNAJC5 has been extensively studied (for instance Greaves et al. 2006 and 2012). Thus, it is perhaps not altogether surprising that DNAJC5 palmitoylation is important for its facilitatory effect on aSyn secretion.

Importance:

The mechanism described in this work appears to mediate the secretion of monomeric, but not aggregated forms (beyond artificially fused tandem repeats of aSyn dimers and tetramers) of synuclein. There is a well-established role for cell-cell transmission of misfolded aSyn oligomers and fibrils in synucleinopathies, which the authors briefly describe in the introduction. However, the role of aSyn monomer secretion in disease (or physiological function) is much less clear. From the context of understanding and treating synucleinopathies, the importance of the findings in this manuscript as currently written are thus somewhat unclear.

Similarly, it would be important to know how generalizable the effects of palmitoylation are on the secretion of other proteins involved in neurodegenerative diseases such as Tau and TDP43, which like aSyn, were previously shown to be dependent on DNAJC5 (Fontaine et al., 2016).

Finally, in this work the authors primarily use a combination of western blot, differential centrifugation, imaging and the introduction of specific mutations to analyze biochemical features of DNAJC5 that contribute to aSyn secretion. While the methods are appropriate, there are limitations to the experimental design (limited technical and biological replication) and choice of model system (predominantly overexpression in a single cellular model) that limit the interpretation of the data presented.

1. A limitation of this work is that the findings appear largely to be drawn from overexpression systems in a single cell line. Only in Figure 1 -Supplement 2 do the authors attempt to determine whether the role of DNAJC5 in regulating synuclein export is conserved under endogenous expression levels, however unfortunately these data are not very convincing. The effect of DNAJC5 knockdown in particular does not appear to support a significant role of the endogenous DNAJC5 protein in α-Syn export (although there may be an effect, the experiment is likely underpowered and no statistically significant effect was reported). The data stemming from the use of quercitin are also difficult to interpret as this natural product has been reported to have multiple targets, especially when used at high doses (the authors here have used it in the micro-molar range). Additional demonstration that DNAJC5 plays a role in α-synuclein export under endogenous expression conditions and in more relevant cellular models would greatly aid the argument that the mechanisms the authors have uncovered are part of a physiologically or disease-relevant pathway.

2. An important limitation of this manuscript is the lack of information regarding replicability of the observations. In many figures only a single western blot showing 1 or 2 replicates are shown. No information is given about whether the replicates that are shown are biological or technical replicates, and no information is given about whether additional experiments were performed (i.e. if these are meant to be representative images or if this represents the sum-total of the data). In figures where quantification has been performed (e.g. Figure 1 supplement 2) error bars are shown with no indication of what they represent (SD versus SEM) or any indication of how many replicates were performed. Similarly, with most immunofluorescence panels only a single image of a single cell is shown, from which the reader is expected to draw conclusions. Showing sufficient replication of the findings (perhaps through quantification performed on the blots and images) would help argue that these data are in fact robust and reproducible. Relatedly, the authors assertion that α-synuclein internalization into endosomes requires a functional DNAJC5 seems to be based largely on visual inspection of a small number of endosomes in single cells. Even among the small number of vesicles shown, it is clear that the amount of α-syn present in vesicles is highly variable, even within a single cell. Therefore, additional (and ideally more quantitative) evidence is required to support the conclusion that endosomal localization requires DNAJC5. In light of these apparent limitations pertaining to both technical and biological replication of some key findings across independent experiments, the data presented are somewhat unconvincing.

Other comments:

Most immunofluorescence figures are inadequately labelled. It is often unclear which images come from which cell lines. It is also recommended to use a color-blind friendly color scheme for all immunofluorescence images.

A methods section for Western blots is absent. This should include the antibodies used and at what dilution. In addition, the antibodies used should be denoted in the Figures or Figure legends for all Western blot panels.

Detailed section-by-section comments:

Introduction

– Several statements in the intro that describe previous findings related to aSyn export or the existence of extracellular aSyn would benefit from specification of the specific state of aSyn that is being referenced (i.e. are the authors speaking about monomeric, fibrillar or other forms of aSyn?)

– Sentence ending on line 65 should have a citation

– In the sentence ending on line 67 the authors refer to "unconventional traffic" but do not provide enough context for the reader to understand what is meant by "unconventional" (i.e. it might help to define conventional and unconventional)

– Line 65 – in previous works showing release of aSyn is mediated by DNAJC5, what form of aSyn was studied?

– Line 71 – please specify under what condition DNAJC5 was found to be neuroprotective.

– In the paragraph starting on line 42 the authors introduce the concept of pathological aSyn spreading and imply a potential role for extracellular secretion, however they later state that DNAJC5 (a protein the reader is then told is involved in this secretion) plays a neuroprotective role (line 71 and 72). Do the authors consider this to be a discrepancy in if so what might explain it? (perhaps a topic to address in the discussion)

Results

– In Figure 1, only secretion of overexpressed aSyn is studied. It might be important to show a role for this pathway in secretion of aSyn at physiological expression levels.

– Line 121: The effect of siRNA knockdown is marginal, and likely statistically insignificant and does not support the authors statements

– Line 150: this statement would be better supported if the authors had a positive control showing that aSyn oligomers can co-ip under the specific conditions used in this experiment (for example can oligomers be co-ip'ed from the cell lysate?)

– Quercitin has been reported to have multiple targets, especially at high doses. Furthermore, the dose of quercitin required to elicit a response in the paper cited by the authors to support the use of this drug is considerably higher than that used by the authors here. Potential off-target effects of the drug should be acknowledged as a limitation.

– In Figure 3 Supplement 1B the relative amount of aSyn to tubulin in cell lysate is not a clear trend. In this figure quantification of the ratio between aSyn and tubulin would assist in interpretation.

Figure 3 D and E – this and other panels showing aSyn in media would benefit from the inclusion of a loading control for EVs ◊ this might help clarify whether a global increase in secretion of EVs might contribute to the increase in aSyn and DNAJC5 observed in media.

– Figure 3 B and C – it appears that although there is a decrease in aSyn secretion following 2-BA treatment, there is no corresponding reduction in cytosolic aSyn. What could explain this discrepancy? Here again, quantifications might assist in uncovering whether small effects are present that are not obvious by visual inspection alone.

– Figure 4 – It appears here that data is shown only either for L115R mutant or WT when in fact these should both be shown side by side.

– In Figure 4 the labelling of which images come from which genetic backgrounds is somewhat unclear. For example, Panel A seems to show staining from cells expressing a wild type DNAJC5 whereas panel B seems to show the L115R mutant, however this is not clearly denoted in the figure or figure legend. It would be helpful if the images corresponding to direct comparisons of the same staining from either WT or mutant were shown side by side in the same figure.

– Figure 4: The conclusion that aSyn internalization into endosomes requires functional DNAJC5 appears to be based only on visual inspection of the 3 representative endosomes shown in panels B and C, which are derived from each from a single cell. In addition, the fractionation experiment is shown only for WT DNAJC5, and not L115R. To reach this conclusion the authors should provide more robust data supporting a difference in endosomal localization of synuclein. This could be achieved by quantifying the aSyn signal in endosomes from a larger number of endosomes, and importantly, from multiple cells derived from distinct biological replicates.

– Figure 4: Confirming the localization of DNAJC5 and aSyn ultrastructurally, for instance by immuno-EM or CLEM would help further support the authors' interpretation regarding localization and topology.

– Figure 4 supplementary video is not referred to at all in the text pertaining to Figure 4, only much later in the discussion.

– Figure 5: It appears that GFP was used as a transfection control in these experiments. In previous Western blot figures no transfection control was indicated. If one was used it should be specified.

– Line 211 should use Greek character for α

– Figure 5 Panel E+F The authors indicate that the asterisk marks a non-specific band, however this does not explain why a non-specific band would appear in only the conditions transfected with the thermostable aSyn mutants. Further, when you compare panels E and F, the location of the non-specific band appears to change relative to aSyn. Interpretation of these western blots would be assisted if molecular weight markers were shown alongside the blots.

– Figure 6: Definition of the abbreviations (PNS, M and C) used in the figure should be provided

– As stated for previous figures, the presence of aSyn in endosomal compartments could more effectively be shown with quantification across multiple replicates

– Figure 7D and Line 311. The blot presented does not provide strong justification for the statement that deletion of the J domain does not affect aSyn secretion. It seems like some mutants may in fact stimulate greater aSyn secretion compared to WT, but it is difficult to draw any conclusion from visual inspection of this blot alone.

---

## [Author Response]

Reviewer #1 (Recommendations for the authors):In this paper, the precise role of DNAJC5 in unconventional secretion of α-synuclein was studied. Elucidating the molecular mechanism of this process is not only an interesting question in basic research but is also of high relevance with regard to potential biomedical applications in the context of neurodegenerative disorders. The current state of the literature is quite heterogenous with a number of distinct pathways being proposed to play a role in α-synuclein secretion, including a potential role for exosomes and cell-connecting nanotubes, among others.Here, the authors aimed at reconstituting α-synuclein secretion in a general human cell line, HEK293T cells. A particular focus was given to DNAJC5, a protein that has previously been demonstrated to play a role in this process. In addition, palmitoylation was already known to mediate membrane attachment of DNAJC5. The authors demonstrate that both palmitoylated and non-palmitoylated DNAJC5 are secreted alongside α-synuclein. However, while palmitoylated DNAJC5 was found associated with some kind of extracellular vesicles, non-palmitoylated DNAJC5 and monomeric α-synuclein were found as soluble proteins in cellular supernatants. The authors further show that palmitoylation of DNAJC5 is required for α-synuclein secretion.Using imaging techniques, the authors found DNAJC5 and α-synuclein to be associated with a population of endosomes. Along with cell fractionation and protease-protection experiments, these experiments were interpreted in a way that both proteins are located in the lumen of endosomes. They were further taken as evidence that DNAJC5 mediates translocation of α-synuclein from the cytoplasm into the lumen of endosomes.In a further set of experiments, the authors aimed at addressing the folding state of α-synuclein while it is transported into the extracellular space. Using an elegant experimental system, evidence was provided that α-synuclein remains folded during all steps of its transport route.Finally, the authors provide evidence for DNAJC5 to oligomerize, a process that is enhanced when the J domain is deleted. They also find that the J domain is not required for α-synuclein secretion.As a bottom line the authors suggest that palmitoylated species of DNAJC5 oligomers form or link to other components of a pore in late endosomes through which α-synuclein translocates into their lumen. This potential step of α-synuclein membrane translocation is proposed to be the critical process that targets α-synuclein into the extracellular space.In general, I appreciate this study that uses classical methods in biochemistry and cell biology collecting a number of important insights about the unconventional secretory pathway of α-synuclein.In the following, I would like to point out a number of questions and suggestions the authors may want to address to improve this manuscript:– Why and how do cells secrete non-palmitoylated DNAJC5, a variant that cannot associate with membranes? And why does palmitoylated DNAJC5 end up on extracellular vesicles, a species that is proposed to be inserted as an oligomer into the limiting membrane of late endosomes? When late endosomes fuse with the plasma membrane as suggested by the authors, palmitoylated DNAJC5 should remain in the inner leaflet of the plasma membrane. Based on the mechanism of α-synuclein secretion proposed in this study, both of these phenomena are difficult to incorporate into the concept of the authors.

The secretion of DNAJC5 in extracellular vesicles has been reported before. In our new data, we also found that endogenous DNAJC5 in differentiated neuroblastoma SH-SY5Y cells was packaged into exosomes, demonstrated both by protease protection assay and sucrose gradient flotation (Figure 7 —figure supplement 1). We think the palmitoylated DNAJC5 (P-DNAJC5) is inserted on the cytosolic side of late endosomes, and then is packaged inside the intraluminal vesicles by membrane invagination, like many other exosome markers, e.g. CD63.

We also were curious about the secretion of non-palmitoylated DNAJC5 (NP-DNAJC5) in HEK293T cells. As the reviewer pointed out, NP- DNAJC5 does not associate with membrane. We noticed that one distinct characteristic of palmitoylation among other protein lipidation is its reversible nature. We have tested if NP-DNAJC5 is generated from depalmitoylation of packaged P-DNAJC5 in late endosomes by knocking out palmitoyl-protein thioesterase 1 (PPT1) in HEK293T cells by CRISPR/Cas9 (Author response image 1). PPT1 has been reported to catalyze the deplamitoylation of P-DNAJC5 in vitro and its deficiency also causes neurodegenerative disorder. However, we did not observe a decrease of secreted NP-DNAJC5 in PPT1 KO cells (Author response image 1). Given several other identified depalmitoylation thioesterases, we think there could be redundant enzymes responsible for the depalmitoylation of DNAJC5. Note we found less NP-DNAJC5 in SH-SY5Y cells (Figure 7A) and expression of a non-reversible XP-DNAJC5 also induced a-syn secretion (Figure 6). Thus we think the P-DNAJC5 is more important for this unconventional secretion pathway and the secretion of NPDNAJC5 may be result from the action of an unknown thioesterase in HEK293T cells.

**Author response image 1. sa2fig1:** 

– I am not sure whether the imaging experiments shown in Figure 4 and its supplements unequivocally demonstrate a luminal localization of the proteins in question. I also find the protease protection experiments to be limited in their conclusiveness since only a minor fraction of α-synuclein and DNAJC5 appear to be protected. Is there a way to produce further independent evidence for these claims? Finally, the time-lapse video provided as a supplement does not convince me with regard to the suggested role of DNAJC5 promoting membrane translocation of α-synuclein.

We performed quantification analysis on the percentage of luminal a-syn with or without DNAJC5. We now report a statistically significant difference of ratio of luminal localized a-syn (Figure 4H).

We acknowledge that the percentage of protected a-syn is low. Given that a-syn is known to bind membrane peripherally, it is not too surprising that much a-syn tethers outside of membrane. We have performed an independent nanoluciferase (Nluc) quenching assay developed by other members in our lab, the principle of which is reported in another manuscript (Williams et al., 2022). Again, the protection efficiency is low (less than 10%) (Author response image 2, left). As a control, cytosolic Nluc-a-syn was not protected. A lithium chloride wash to reduce peripherally associated a-syn increased the protected percentage to ~25% (Author response image 2, right).

– To me it seems unclear whether the populations of α-synuclein and DNAJC5 found on endosomes are necessarily linked to the unconventional secretory mechanism of α-synuclein. Isn't it also possible that this material is derived from endocytosis of α-synuclein and DNAJC5? In particular with regard to the populations that are potentially in the lumen of these endosomes?

The visualization of proteins in the lumen of enlarged multivesicular bodies (MVBs) was widely used to indicate protein packaged into exosomes. We repeated the experiments by treating the cells with 80 μm dynasore to inhibit endocytosis. A-syn and DNAJC5 are still found in the lumen of the endosomes (Author response image 3).

**Author response image 3. sa2fig3:** 

– The experimental evidence for DNAJC5 oligomers mediating membrane translocation of α-synuclein remains limited. I think this paper could contribute a major advance to the field if the observed oligomers could be characterized in more detail as has been done in other studies in the field. For example, beyond the observation of SDS-resistant oligomers in gels, FCS techniques could be used to precisely determine the oligomeric state of DNAJC5 in a native state. In addition. there are biophysical assays that allow to observe pore formation with purified proteins and artificial membrane vesicles. If there is a pore, its cut-off size could be determined and compared to the dimensions of α-synuclein. The authors indicate in the discussion that DNAJC5 might not be able to form a pore itself and that this process might be coupled to CD98. I think these aspects remain vague, however, insights into these questions would make up a true advance in the field.

We agree that the insights into the oligomerization of DNAJC5 and identification of other components in the translocation machineries will be the important next step in our research. However, we don’t have access to FCS in the lab and the biophysical analysis of in vitro reconstituted DNAJC5 oligomers is challenging. Our gel filtration results showed that DNAJC5 formed oligomers in a non-denaturing environment (without SDS). Currently, we are working on identification of DNAJC5 interacting proteins for a-syn translocation. This will be covered in our future work.

– While the technique used to study the folding state of α-synuclein during its transport into the extracellular space is very elegant, in my opinion a positive control is missing, i.e. an unconventionally secreted protein whose secretion depends on an intermediate state that is unfolded. I think Interleukin 1β could serve as such a control.

We thank the reviewer to suggest this critical control. After we consulted with the authors of Interleukin 1b (IL1b) secretion, they raised the concern that 3H bundle may alter the membrane localization of IL1b based on its sequence. Instead, we turned to the mitochondrial protein import pathway, which is also dependent on protein-unfolding. This control has been added as a supplementary figure where the incorporation of 3H after the transit peptide blocked import of a mitochondrial matrix reporter protein (Figure 5 —figure supplement 2).

Reviewer #2 (Recommendations for the authors):In this manuscript, Wu et al. investigate the mechanisms through which the co-chaperone DNAJC5 regulates unconventional secretion of α-synuclein. The authors have some interesting findings such as palmitoylation is important for this process, but whether DNAJC5 itself was necessary is unclear. Addressing this ambiguity could lead to meaningful insights and the study could provide a cell biological basis for α-synuclein spread in Parkinson's disease.Several additional experiments and quantification of the data presented will greatly enhance the quality of this manuscript.Figure 1-NP-DNAJC5 and P-DNAJC5. The palmitoylation status of P-DNAJC5 should be confirmed by treating with hydroxylamine and checking if the protein now runs at NP-DNAJC5.

The hydroxylamine experiment was performed and added as a supplementary figure (Figure 1—figure supplement 1).

Figure 1-Please rule out that expression of the various constructs is not causing cell death.

The cell viability assay was performed and added as a supplementary figure (Figure 1—figure supplement 2A).

Figure 1-S1-Can the authors comment on why DNAJC5 levels in the medium is decreased when PD mutants of α-synuclein are expressed? It would be helpful to quantitate these data to see the impact of PD mutations.

This quantification was performed and is labeled in the figures. The a-syn PD mutants are known to affect its aggregation propensity. It is possible that the DNAJC5 co-aggregates with a-syn PD mutants so that less DNAJC5 is secreted. Further experiments would need to be performed to address such a hypothesis but that is beyond the scope of the current manuscript.

Figure 1-S2-Quercetin is rather non-specific and toxic. The data would be bolstered if in Figure S2E, the luciferase data is accompanied by western blots for α-synuclein and DNAJC5 (in addition to tubulin).

The immunoblots of a-syn and DNAJC5 were added in a supplementary figure (Figure 1—figure supplement 3E).

Figure 1-S2F-G- The effect of the KD should be quantified. KD of DNJAC5 does not seem to impact α-synuclein secretion. Is the effect on α-synuclein secretion even significant? These data with Figure 3 would argue that DNAJC5 itself is not necessary for secretion of α-synuclein. In this regard, can you substitute DNAJC5 with any heavily palmitoylated protein?

We generated a DNAJC5 CRISPR/Cas9 KO HEK293T cell line and repeated the experiment. The effect of DNAJC5 is more significant with the treatment of BaFA1. We have updated the supplementary figure with the KO results accordingly.

Figure 1-S2F-G- These data suggest that PATs that palmitoylate DNAJC may be important for secretion of α-synuclein. Also, KD of HIP14 would be instructive.

We tried to generate CRIPSR/Cas9 HIP14 KO HEK293T cells. Unfortunately, we could not find a good antibody against HIP14 in HEK293T cells but we found decreased P-DNAJC5 in several clones. Accordingly, the level of extracellular a-syn decreased about 40%.

**Author response image 4. sa2fig4:** 

Figure 3-the ratio of DNAJC5-NP/P for the various conditions should be quantified.

The ratio has been quantified and added as a new Panel to the Figure 3 (Figure 3B).

Figure 3-The L115R mutant is retained in the Golgi. Does treatment with 2-BA also result in retention of DNAJC5 in the Golgi? If so, how does this fit into the model proposed for α-synuclein secretion.

We confirmed Golgi retention of the L115R mutant protein by immunofluorescence (IF). As suggested by the reviewer, 2-BA treatment also resulted in Golgi retention of DNAJC5. Based on the observation above, we think that mislocalized NP-DNAJC5 failed to insert into the membrane of late endosomes, thus as for soluble DNAJC5 in the cytosol, was unable to mediate a-syn secretion. The Golgi retention results are summarized in Figure 3 —figure supplement 3.

Figure 4-How many large endosomes contain both DNAJC5 and α-synuclein? This is important to ascertain as it is not evident from the images shown that the miniscule amount of α-synuclein in endosomes accounts for amounts seen by western blotting of media. Also, why is the α-synuclein within the endosome not diffuse (as would be expected from a monomer) but rather punctate?

We quantified the DNAJC5 and a-synuclein containing endosomes and the difference between the absence and presence of DNAJC5 is significant (Figure 4H). The miniscule amount of internalized asynuclein is contributed by the high background of cytosolic a-syn.

In respect to the punctate distribution of internalized a-syn, we think that it is possible that soluble asyn may tether on the membrane of intraluminal vesicles. The stickiness of a-syn may contribute to membrane-association inside the lumen of endosomes.

Figure 4-mNeo-Green tagging in the N-terminus of α-synuclein will prevent the protein from folding properly.

As we demonstrate in Figure 5, DNAJC5-mediated a-syn secretion is independent of protein unfolding. The secretion of N-terminally tagged a-syn is confirmed as normal (Author response image 5). We also generated an a-syn construct with C-terminally tagged mNeon-Green (a-syn-mNG). The internalization of C-tagged a-syn-mNG was also confirmed by live cell imaging (Author response image 5).

**Author response image 5. sa2fig5:** 

Lastly recapitulating these data in neurons would increase the significance of these findings.

We have recapitulated several key data in RA treated differentiated neuroblastoma (SH-SY5Y) cells, in human patient-derived iPSC differentiated dopaminergic neurons and in mouse mESCdifferentiated dopaminergic neurons. All these new data are summarized in an updated Figure 7.

Discussion is very speculative.

We have updated Discussion section based on our new data.

Reviewer #3 (Recommendations for the authors):The authors describe a mechanism by which DNAJC5 stimulates the extracellular secretion of α-synuclein in a manner dependent on the palmitoylation of DNAJC5 which serves to anchor the protein to the membrane. There may be an additional role for oligomerization of DNAJC5 in facilitating α-syn secretion, as modifications to the protein which induce oligomerization appear to rescue deficits observed in palmitoyl-deficient mutants. The authors propose that the secretion of α-synuclein proceeds through an intermediate step of localization into late endosomal compartments, which appears to be disrupted in DNAJC5 mutants. Broadly this paper uses a combination of cellular fractionation, immunoblot and immunofluorescence microscopy in HEK 293 cells overexpressing a range of DNAJC5 constructs to dissect the role of this protein in α-synuclein export.Advance over previous work:The observation that DNAJC5 can mediate extracellular secretion of aSyn was previously described (Fontaine et al., 2016). Similarly, the importance of palmitoylation of DNAJC5 has been extensively studied (for instance Greaves et al. 2006 and 2012). Thus, it is perhaps not altogether surprising that DNAJC5 palmitoylation is important for its facilitatory effect on aSyn secretion.Importance:The mechanism described in this work appears to mediate the secretion of monomeric, but not aggregated forms (beyond artificially fused tandem repeats of aSyn dimers and tetramers) of synuclein. There is a well-established role for cell-cell transmission of misfolded aSyn oligomers and fibrils in synucleinopathies, which the authors briefly describe in the introduction. However, the role of aSyn monomer secretion in disease (or physiological function) is much less clear. From the context of understanding and treating synucleinopathies, the importance of the findings in this manuscript as currently written are thus somewhat unclear.

We consistently see the majority of extracellular a-syn secreted is soluble, both in HEK293T cells and in neuronal cells. Previous publications also reported that increased extracellular a-syn in patient-derived neuronal culture medium is soluble and not associated with exosomes (Fernandes et al., Stem Cell Reports, 2016). The release of soluble a-syn can be a protective mechanism to alleviate the intracellular a-syn pathology before the onset of synucleinopathies. We revised the discussion to clarify the relevance of our findings.

Similarly, it would be important to know how generalizable the effects of palmitoylation are on the secretion of other proteins involved in neurodegenerative diseases such as Tau and TDP43, which like aSyn, were previously shown to be dependent on DNAJC5 (Fontaine et al., 2016).

We focus on the mechanism of a-syn secretion in this manuscript but a visiting scholar in our lab has been working on DNAJC5 regulated Tau secretion and confirmed some of our findings in Tau secretion. This will be covered in their research.

Finally, in this work the authors primarily use a combination of western blot, differential centrifugation, imaging and the introduction of specific mutations to analyze biochemical features of DNAJC5 that contribute to aSyn secretion. While the methods are appropriate, there are limitations to the experimental design (limited technical and biological replication) and choice of model system (predominantly overexpression in a single cellular model) that limit the interpretation of the data presented.

We have quantified our imaging and blots, and recapitulated our findings in several neuronal cell culture lines at endogenous level.

1. A limitation of this work is that the findings appear largely to be drawn from overexpression systems in a single cell line. Only in Figure 1 -Supplement 2 do the authors attempt to determine whether the role of DNAJC5 in regulating synuclein export is conserved under endogenous expression levels, however unfortunately these data are not very convincing. The effect of DNAJC5 knockdown in particular does not appear to support a significant role of the endogenous DNAJC5 protein in α-Syn export (although there may be an effect, the experiment is likely underpowered and no statistically significant effect was reported). The data stemming from the use of quercitin are also difficult to interpret as this natural product has been reported to have multiple targets, especially when used at high doses (the authors here have used it in the micro-molar range). Additional demonstration that DNAJC5 plays a role in α-synuclein export under endogenous expression conditions and in more relevant cellular models would greatly aid the argument that the mechanisms the authors have uncovered are part of a physiologically or disease-relevant pathway.

We generated a CRISPR/Cas9 DNAJC5 KO HEK293T cells and repeated the experiment. The effect of DNAJC5 is more significant with the treatment of BaFA1. We have updated the supplementary figure with the KO results accordingly.

We have recapitulated several key data in differentiated neuroblastoma cells, human patient-derived iPSC differentiated dopaminergic neurons and mouse mESC differentiate dopaminergic neurons. Please see our updated Figure 7.

2. An important limitation of this manuscript is the lack of information regarding replicability of the observations. In many figures only a single western blot showing 1 or 2 replicates are shown. No information is given about whether the replicates that are shown are biological or technical replicates, and no information is given about whether additional experiments were performed (i.e. if these are meant to be representative images or if this represents the sum-total of the data). In figures where quantification has been performed (e.g. Figure 1 supplement 2) error bars are shown with no indication of what they represent (SD versus SEM) or any indication of how many replicates were performed. Similarly, with most immunofluorescence panels only a single image of a single cell is shown, from which the reader is expected to draw conclusions. Showing sufficient replication of the findings (perhaps through quantification performed on the blots and images) would help argue that these data are in fact robust and reproducible. Relatedly, the authors assertion that α-synuclein internalization into endosomes requires a functional DNAJC5 seems to be based largely on visual inspection of a small number of endosomes in single cells. Even among the small number of vesicles shown, it is clear that the amount of α-syn present in vesicles is highly variable, even within a single cell. Therefore, additional (and ideally more quantitative) evidence is required to support the conclusion that endosomal localization requires DNAJC5. In light of these apparent limitations pertaining to both technical and biological replication of some key findings across independent experiments, the data presented are somewhat unconvincing.

We have added the information when applicable in our figure legends and performed the quantification of endosomes.

Other comments:Most immunofluorescence figures are inadequately labelled. It is often unclear which images come from which cell lines. It is also recommended to use a color-blind friendly color scheme for all immunofluorescence images.

We have added cell line information in the figure legends.

A methods section for Western blots is absent. This should include the antibodies used and at what dilution. In addition, the antibodies used should be denoted in the Figures or Figure legends for all Western blot panels.

We have added an immunoblot section in Methods. The antibodies used were listed in Key resource table. The dilution number was summarized in the method section.

Detailed section-by-section comments:Introduction– Several statements in the intro that describe previous findings related to aSyn export or the existence of extracellular aSyn would benefit from specification of the specific state of aSyn that is being referenced (i.e. are the authors speaking about monomeric, fibrillar or other forms of aSyn?)

We clarify the state of extracellular a-syn (monomers, oligomers and aggregates) in the Introduction.

– Sentence ending on line 65 should have a citation

The citation has been added (Fontaine et al., 2016).

– In the sentence ending on line 67 the authors refer to "unconventional traffic" but do not provide enough context for the reader to understand what is meant by "unconventional" (i.e. it might help to define conventional and unconventional)

The definition of “unconventional secretion” has been added.

– Line 65 – in previous works showing release of aSyn is mediated by DNAJC5, what form of aSyn was studied?

In Fontaine et al., 2016, the form of a-syn was not specified.

– Line 71 – please specify under what condition DNAJC5 was found to be neuroprotective.

Conventionally DNAJC5 is considered as a chaperone to prevent protein misfolding. Deletion and mutations in DNAJC5 cause neurodegeneration and premature death. The conditions were added in the text.

– In the paragraph starting on line 42 the authors introduce the concept of pathological aSyn spreading and imply a potential role for extracellular secretion, however they later state that DNAJC5 (a protein the reader is then told is involved in this secretion) plays a neuroprotective role (line 71 and 72). Do the authors consider this to be a discrepancy in if so what might explain it? (perhaps a topic to address in the discussion)

Calo et al., 2021 showed that DNAJC5 was able to reduce a-syn aggregates and increase monomeric a-syn. We also showed that the secreted a-syn mediated by DNAJC5 is mainly soluble with no stress conditions. In addition to its chaperone function, we hypothesized that DNAJC5 may play a positive role in helping to dispose of a-syn by secretion to prevent aggregation. We added a new paragraph in our discussion to address this.

Results– In Figure 1, only secretion of overexpressed aSyn is studied. It might be important to show a role for this pathway in secretion of aSyn at physiological expression levels.

We studied secretion of endogenous a-syn in several neuronal cultures. In these neuronal cultures, DNAJC5 also plays a role in the secretion of soluble a-syn.

– Line 121: The effect of siRNA knockdown is marginal, and likely statistically insignificant and does not support the authors statements

We have generated a HEK293T DNAJC5 CRIPSR knockout (KO) cell line. BaFA1-stimulated a-syn secretion was decreased in DNAJC5 KO cell line.

– Line 150: this statement would be better supported if the authors had a positive control showing that aSyn oligomers can co-ip under the specific conditions used in this experiment (for example can oligomers be co-ip'ed from the cell lysate?)

The formation of a-syn oligomers is challenging in the cell culture conditions we used. When examined by gel filtration, almost all a-syn in cell lysate were soluble. Co-immunoprecipitation of oligomerized protein complexes has been widely used to detect the interaction between protomers.

– Quercitin has been reported to have multiple targets, especially at high doses. Furthermore, the dose of quercitin required to elicit a response in the paper cited by the authors to support the use of this drug is considerably higher than that used by the authors here. Potential off-target effects of the drug should be acknowledged as a limitation.

The titration experiment is used to optimize the concentration to inhibit DNAJC5 palmitoylation. Tubulin was used here only to monitor cell toxicity in the case of released tubulin in the medium, which was not observed in the experiment. We quantified the ratio of a-syn in media / a-syn in lysates to represent normalized a-syn secretion. The inhibition effect of DNAJC5 palmitoylation was quantified by the ratio between P-DNAJC5 and NP-DNAJC5 in the cell lysate.

We acknowledged the potential off-target effect of quercetin in the main text.

– In Figure 3 Supplement 1B the relative amount of aSyn to tubulin in cell lysate is not a clear trend. In this figure quantification of the ratio between aSyn and tubulin would assist in interpretation.

The titration experiment is used to optimize the concentration to inhibit DNAJC5 palmitoylation. Tubulin was used here only to monitor cell toxicity in the case of released tubulin in the medium, which was not observed in the experiment. We quantified the ratio of a-syn in media / a-syn in lysates to represent normalized a-syn secretion. The inhibition effect of DNAJC5 palmitoylation was quantified by the ratio between P-DNAJC5 and NP-DNAJC5 in the cell lysate.

Figure 3 D and E – this and other panels showing aSyn in media would benefit from the inclusion of a loading control for EVs ◊ this might help clarify whether a global increase in secretion of EVs might contribute to the increase in aSyn and DNAJC5 observed in media.

The detection of EV in this experiment is challenging, since all the secretion assays were performed with cell culture in 6-well plates. Detection of EV markers typically require large-scale culture.

– Figure 3 B and C – it appears that although there is a decrease in aSyn secretion following 2-BA treatment, there is no corresponding reduction in cytosolic aSyn. What could explain this discrepancy? Here again, quantifications might assist in uncovering whether small effects are present that are not obvious by visual inspection alone.

The population of secreted a-syn is only minor fraction of total a-syn in cell. We quantify the influence of a-syn secretion by both 2-BA treatment and L115R mutation.

In figure 4, we focus on the translocation of WT DNAJC5. The effect of L115R and rescue by XPL115R was shown in Figure 6.

– Figure 4 – It appears here that data is shown only either for L115R mutant or WT when in fact these should both be shown side by side.

In figure 4, we focus on the translocation of WT DNAJC5. The effect of L115R and rescue by XPL115R was shown in Figure 6.

– In Figure 4 the labelling of which images come from which genetic backgrounds is somewhat unclear. For example, Panel A seems to show staining from cells expressing a wild type DNAJC5 whereas panel B seems to show the L115R mutant, however this is not clearly denoted in the figure or figure legend. It would be helpful if the images corresponding to direct comparisons of the same staining from either WT or mutant were shown side by side in the same figure.

All the images shown in Figure 4 main figures are from DNAJC5 WT. Panel B show the localization of a-syn, not DNAJC5 L115R mutant. The names of each channel were labeled in images.

– Figure 4: The conclusion that aSyn internalization into endosomes requires functional DNAJC5 appears to be based only on visual inspection of the 3 representative endosomes shown in panels B and C, which are derived from each from a single cell. In addition, the fractionation experiment is shown only for WT DNAJC5, and not L115R. To reach this conclusion the authors should provide more robust data supporting a difference in endosomal localization of synuclein. This could be achieved by quantifying the aSyn signal in endosomes from a larger number of endosomes, and importantly, from multiple cells derived from distinct biological replicates.

We quantified the ratio of a-syn containing endosomes from multiple cells. A statistically significant difference was found between a-syn alone (Control) and a-syn co-transfected with DNAJC5 (Figure 4H).

– Figure 4: Confirming the localization of DNAJC5 and aSyn ultrastructurally, for instance by immuno-EM or CLEM would help further support the authors' interpretation regarding localization and topology.

The CLEM is challenging because most a-syn is soluble. Instead, a protease protection assay was performed to cross-validate our imaging data (Figure 4Fand4G).

– Figure 4 supplementary video is not referred to at all in the text pertaining to Figure 4, only much later in the discussion.– Figure 5: It appears that GFP was used as a transfection control in these experiments. In previous Western blot figures no transfection control was indicated. If one was used it should be specified.– Line 211 should use Greek character for α– Figure 5 Panel E+F The authors indicate that the asterisk marks a non-specific band, however this does not explain why a non-specific band would appear in only the conditions transfected with the thermostable aSyn mutants. Further, when you compare panels E and F, the location of the non-specific band appears to change relative to aSyn. Interpretation of these western blots would be assisted if molecular weight markers were shown alongside the blots.– Figure 6: Definition of the abbreviations (PNS, M and C) used in the figure should be provided– As stated for previous figures, the presence of aSyn in endosomal compartments could more effectively be shown with quantification across multiple replicates– Figure 7D and Line 311. The blot presented does not provide strong justification for the statement that deletion of the J domain does not affect aSyn secretion. It seems like some mutants may in fact stimulate greater aSyn secretion compared to WT, but it is difficult to draw any conclusion from visual inspection of this blot alone.